# Statistical Error Bounds for GANs with Nonlinear Objective Functionals

**Jeremiah Birrell**
*Department of Mathematics*
*Texas State University*
*San Marcos, TX 78666, USA*

*jbirrell@txstate.edu*

**Reviewed on OpenReview:** *https://openreview.net/forum?id=ZgjhykPSdU*

## Abstract

Generative adversarial networks (GANs) are unsupervised learning methods for training a generator distribution to produce samples that approximate those drawn from a target distribution. Many such methods can be formulated as minimization of a metric or divergence between probability distributions. Recent works have derived statistical error bounds for GANs that are based on integral probability metrics (IPMs), e.g., WGAN which is based on the 1-Wasserstein metric. In general, IPMs are defined by optimizing a linear functional (difference of expectations) over a space of discriminators. A much larger class of GANs, which we here call $(f, \Gamma)$-GANs, can be constructed using $f$-divergences (e.g., Jensen-Shannon, KL, or $\alpha$-divergences) together with a regularizing discriminator space $\Gamma$ (e.g., 1-Lipschitz functions). These GANs have nonlinear objective functions, depending on the choice of $f$, and have been shown to exhibit improved performance in a number of applications. In this work we derive statistical error bounds for $(f, \Gamma)$-GANs for general classes of $f$ and $\Gamma$ in the form of finite-sample concentration inequalities. These results prove the statistical consistency of $(f, \Gamma)$-GANs and reduce to the known results for IPM-GANs in the appropriate limit. Our results use novel Rademacher complexity bounds which provide new insight into the performance of IPM-GANs for distributions with unbounded support and have application to statistical learning tasks beyond GANs.

## 1 Introduction

Generative adversarial networks (GANs) are unsupervised learning methods for training a generator distribution to approximate a target distribution by using samples from the target in a minmax game between a generator and a discriminator network (Goodfellow et al., 2014; Dziugaite et al., 2015; Li et al., 2015; Arjovsky et al., 2017). Mathematically, many such methods can be formulated in terms of minimizing an integral probability metric (IPM) :

$$\inf_{\theta \in \Theta} d_\Gamma(Q, P_\theta) \,, \tag{1}$$

$$d_\Gamma(Q, P_\theta) \coloneqq \sup_{h \in \Gamma} \{E_Q[h] - E_{P_\theta}[h]\} \,. \tag{2}$$

Here $d_\Gamma$ is the IPM with test function space $\Gamma$ (i.e., discriminators), $Q$ is the distribution to be learned (i.e., the distribution of the data) and $P_\theta$ is the generator distribution, depending on parameters $\theta \in \Theta$. More specifically, $P_\theta = (\Phi_\theta)_\# P_Z$ is the pushforward of a noise source $P_Z$ under a generator network $\Phi_\theta$. For instance, Wasserstein GAN (Arjovsky et al., 2017) corresponds to $\Gamma$ being the set of 1-Lipschitz functions. A key distinguishing feature of the IPM-GANs is the linearity in $h$ of the objective functional in (2).

A larger class of GAN methods can be formulated by generalizing the minmax game (1) to use a nonlinear objective functional; such methods can generally be viewed as minimizing a divergence (a generalized notion

of "distance" or discrepancy) between $Q$ and $P_\theta$. Such methods include the original GAN (Goodfellow et al., 2014) which was based on the Jensen-Shannon (JS) divergence, the more general $d_{\mathcal{F},\phi}$-divergences defined in Arora et al. (2017), which use nonlinear objective functionals that generalize the original JS-GAN objective by replacing the log with another concave function $\phi$, the nonlinear objectives $f(x,y)$ introduced in Liu et al. (2017), the convex duality framework of Farnia & Tse (2018), and the restricted $f$-divergences of Liu & Chaudhuri (2018). In this paper we will focus on GANs expressed in terms of $(f,\Gamma)$-divergences; these divergences generalize and interpolate between IPMs and $f$-divergences (e.g., KL-divergence) and the corresponding GANs have nonlinear objectives; see Birrell et al. (2022b;a) for the general theory of these objects which, in contrast to a number of other approaches, applies to probability distributions having unbounded support. The corresponding $(f,\Gamma)$-GANs encompass and generalize a large number of successful methods in the literature (Goodfellow et al., 2014; Nowozin et al., 2016; Belghazi et al., 2018; Miyato et al., 2018; Arjovsky et al., 2017; Gulrajani et al., 2017; Song & Ermon, 2020a; Nguyen et al., 2010; Gretton et al., 2012; Glaser et al., 2021; Dupuis, Paul & Mao, Yixiang, 2022); see Table 2 in Birrell et al. (2022b). Moreover, we are further motivated to study the statistical properties of $(f,\Gamma)$-GANs due to the observation that the nonlinearity of the objective functional confers several advantages over Wasserstein GAN (WGAN), a widely-used IPM GAN. Specifically, in Birrell et al. (2022b), $(f,\Gamma)$-GANs were shown to be effective on heavy-tailed data for which WGAN fails. $(f,\Gamma)$-GANs were also shown in experiments to be significantly less sensitive to hyperparameter tuning, with $(f,\Gamma)$-GAN training being successful under hyperparameter perturbations which caused WGAN training to fail. Those experiments focused on $f$'s coming from the family $\alpha$-divergence; see Table 1 below.

The $(f,\Gamma)$-GAN are constructed from the $(f,\Gamma)$-divergence, which are defined as follows. Given a convex function $f$ with $f(1)=0$ and a test function space $\Gamma$, the $(f,\Gamma)$-divergence between probability distributions $Q$ and $P$ is defined by

$$D_f^\Gamma(Q\|P) \coloneqq \sup_{h\in\Gamma}\left\{E_Q[h] - \Lambda_f^P[h]\right\}, \tag{3}$$

where, denoting the Legendre transform of $f$ by $f^*$, we define

$$\Lambda_f^P[h] \coloneqq \inf_{\nu\in\mathbb{R}}\{\nu + E_P[f^*(h-\nu)]\}. \tag{4}$$

We refer to $\Lambda_f^P$ as the generalized cumulant generating function because in the KL-divergence case ($f_{KL}(z) = z\log(z)$) it is straightforward to show $\Lambda_{f_{KL}}^P[h] = \log E_P[e^g]$, which is the classical cumulant generating function. The relation of $\Lambda_f^P$ to $f$-divergences was previously studied in Broniatowski & Keziou (2006); Ben-Tal & Teboulle (2007); Nguyen et al. (2010); Ruderman et al. (2012). Under appropriate assumptions on $f$ and $\Gamma$, $D_f^\Gamma(Q\|P)$ provides a meaningful notion of discrepancy between $Q$ and $P$ due to it satisfying the divergence property, i.e., $D_f^\Gamma(Q\|P) \geq 0$ with equality if and only if $Q = P$; see Theorem 2.8 in Birrell et al. (2022b). This property, along with the variational characterization (3), motivates the definition of $(f,\Gamma)$-GANs:

$$\inf_{\theta\in\Theta} D_f^\Gamma(Q\|P_\theta) = \inf_{\theta\in\Theta}\sup_{h\in\Gamma}\left\{E_Q[h] - \Lambda_f^P[h]\right\}. \tag{5}$$

We note that (3) can alternatively be written as

$$D_f^\Gamma(Q\|P) \coloneqq \sup_{h\in\widetilde{\Gamma}}\left\{E_Q[h] - E_P[f^*(h)]\right\}, \tag{6}$$

with the shifts $\nu\in\mathbb{R}$ absorbed into the definition of the discriminator space $\widetilde{\Gamma} \coloneqq \{h-\nu : h\in\Gamma, \nu\in\mathbb{R}\}$ (i.e., $\nu$ is the bias parameter of the final layer). The form (6) is preferable for implementation purposes, however for the purposes of our analysis we will focus on the form (3) in order to emphasize the special role that the parameter $\nu$ plays. More specifically, our methods for proving statistical consistency will rely on the assumption that $\Gamma$ consists of uniformly bounded functions, while the a priori unbounded shift parameter $\nu$ will require special attention. Such considerations are not relevant for IPM GANs since a bias parameter in the final layer will exactly cancel due to the linearity of the objective functions, and hence can be neglected.

The statistical consistency theory of IPM-GANs has recently been studied in a number of works (Liang, 2021; Biau et al., 2021; Huang et al., 2022; Chen et al., 2023b; Chakraborty & Bartlett, 2024). These derivations take advantage of the special structure of IPMs, namely the linearity of the objective functional. As previously noted, a number of works have introduced GANs with nonlinear objective, however the statistical consistency theory for such GANs is largely lacking. A notable exception is Arora et al. (2017) which studied the original JS-GAN and their generalization to $d_{\mathcal{F},\phi}$-divergences; we will provide a detailed comparison of this earlier work with ours in Section 1.2 below. In this paper we develop theory that allows us to prove statistical consistency of the $(f,\Gamma)$-GANs, in the form of finite-sample concentration inequalities. The key technical hurdle is the nonlinearity of the objective functional due to the presence of the generalized cumulant generating function (4). Section 2 is dedicated to properties of and bounds on $\Lambda_f^P$ which will be needed in Section 3 to derive concentration inequalities for $(f,\Gamma)$-GANs. We note that our theory does not make any compactness assumptions on the support of the data distribution or generator noise source; our method for handling distributions with unbounded support in Section 3.3 provides new insight even in the IPM-GAN case.

## 1.1 Summary of Results

Key to our results is a decomposition of the $(f,\Gamma)$-GAN error into optimization error, approximation error, and various statistical errors. Given an approximate discriminator space $\widetilde{\Gamma} \subset \Gamma$ (e.g., a neural network with spectral normalization as an approximation to the space of 1-Lipschitz functions), and empirical measures $Q_n$ and $P_{\theta,m} := (\Phi_\theta)_\# P_{Z,m}$ constructed using $n$ i.i.d. samples from the data distribution, $Q$, and $m$ i.i.d. samples from the generator noise source, $P_Z$, respectively, we let $\theta_{n,m}^*$ denote a solution (with optimization error tolerance $\epsilon_{opt}^{n,m} \geq 0$) to the empirical $(f,\widetilde{\Gamma})$-GAN problem:

$$D_f^{\widetilde{\Gamma}}(Q_n \| P_{\theta_{n,m}^*,m}) \leq \inf_{\theta \in \Theta} D_f^{\widetilde{\Gamma}}(Q_n \| P_{\theta,m}) + \epsilon_{opt}^{n,m}. \tag{7}$$

Given this, we derive the following $(f,\Gamma)$-GAN error bound in Lemma 3.4:

$$D_f^\Gamma(Q \| P_{\theta_{n,m}^*,m}) - \inf_{\theta \in \Theta} D_f^\Gamma(Q \| P_\theta) \tag{8}$$

$$\leq \underbrace{\sup_{h \in \widetilde{\Gamma}, \theta \in \Theta} \left\{ \Lambda_f^{P_{Z,m}}[h \circ \Phi_\theta] - \Lambda_f^{P_Z}[h \circ \Phi_\theta]) \right\} + \sup_{h \in \widetilde{\Gamma}, \theta \in \Theta} \left\{ \Lambda_f^{P_Z}[h \circ \Phi_\theta] - \Lambda_f^{P_{Z,m}}[h \circ \Phi_\theta] \right\}}_{\text{statistical error from sampling } P_Z}$$

$$+ \underbrace{\sup_{h \in \widetilde{\Gamma}} \left\{ E_Q[h] - E_{Q_n}[h] \right\} + \sup_{h \in \widetilde{\Gamma}} \left\{ E_{Q_n}[h] - E_Q[h] \right\}}_{\text{statistical error from sampling } Q}$$

$$+ \underbrace{\epsilon_{n,m}^{opt}}_{\text{optimization error}} + \underbrace{\left(1 + (f^*)_+'(z_0 + \beta - \alpha)\right) \sup_{h \in \Gamma} \inf_{\tilde{h} \in \widetilde{\Gamma}} \|h - \tilde{h}\|_\infty}_{\text{discriminator approximation error}}.$$

The distribution $P_{\theta_{n,m}^*,m} = (\Phi_{\theta_{n,m}^*})_\# P_Z$ is the $(f,\Gamma)$-GAN estimator, i.e., the approximation to $Q$ obtained by solving the empirical GAN problem (7). The primary difference between the decomposition (8) and the corresponding result for IPMs in Lemma 9 of Huang et al. (2022) is the presence of the generalized cumulant generating function $\Lambda_f^{P_Z}$ in the statistical error from sampling $P_Z$. The functional $\Lambda_f^{P_Z}$ is nonlinear in the discriminator and so treating those terms requires new techniques which we develop in Section 2. Using these new results, in the case of a discriminator space $\Gamma$ such that $\sup_{x,\tilde{x}} |h(x) - h(\tilde{x})| \leq 1$ for all $h \in \Gamma$, we derive concentration inequalities of the following form; see Theorem 3.7 for details:

$$\mathbb{P}\left(D_f^\Gamma(Q \| P_{\theta_{n,m}^*,m}) - \inf_{\theta \in \Theta} D_f^\Gamma(Q \| P_\theta) \geq \epsilon + \epsilon_{approx}^{\Gamma,\widetilde{\Gamma}} + \epsilon_{opt}^{n,m} + 4\mathcal{R}_{\widetilde{\Gamma},Q,n} + 4\mathcal{K}_{f,\widetilde{\Gamma}\circ\Phi,P_Z,m}\right) \leq \exp\left(-\frac{\epsilon^2}{\frac{2}{n} + \frac{2}{m}\Delta_{f,m}^2}\right). \tag{9}$$

Here $\epsilon_{approx}^{\Gamma,\widetilde{\Gamma}}$ is the discriminator approximation error, $\mathcal{R}_{\widetilde{\Gamma},Q,n}$ denotes the Rademacher complexity of $\widetilde{\Gamma}$ (see Appendix C), $\mathcal{K}_{f,\widetilde{\Gamma}\circ\Phi,P_Z,m}$ depends on $\mathcal{R}_{\widetilde{\Gamma}\circ\Phi,P_Z,m}$, and $\Delta_{f,m}$ can be bounded uniformly in $m$. We note that

one can always choose $\Gamma = \widetilde{\Gamma}$, thereby making the discriminator approximation error zero; in general, $\epsilon^{\Gamma, \widetilde{\Gamma}}_{approx}$ simply quantifies the tradeoff between using a weaker discriminator space $\widetilde{\Gamma}$ in the Rademacher complexity terms, as compared to the stronger space $\Gamma$ used in the $(f, \Gamma)$-divergence. The approximation error has already been studied for IPM GANs and, apart from the $f^*$ dependence of the prefactor, there is no difference here. Hence we refer the reader to Section 2.2.2 in Huang et al. (2022) for further discussion of this term.

Assuming the discriminator and optimization errors are zero and assuming that one chooses discriminator and generator classes whose Rademacher complexities decay as $n, m \to \infty$, our result (9) shows that the $(f, \Gamma)$-GAN estimator $P_{\theta^*_{n,m}}$ approximately solves the exact $(f, \Gamma)$-GAN problem (5) with high probability. If the generator class is sufficiently rich to ensure $\inf_{\theta \in \Theta} D_f^{\Gamma}(Q \| P_\theta) = 0$ then (9) implies the $(f, \Gamma)$-GAN estimator is close to the distribution of the data source, $Q$, with high probability, where closeness is measured by the $(f, \Gamma)$-divergence; in this latter scenario we in fact obtain a tighter result than (9), see (35). The exponentially decaying nature of the bounds (9) means that one can use the Borel-Cantelli lemma to also conclude almost-sure convergence of $D_f^{\Gamma}(Q \| P_{\theta^*_{n,m}})$ to $\inf_{\theta \in \Theta} D_f^{\Gamma}(Q \| P_\theta)$, provided that the errors and Rademacher complexities approach zero when $n, m \geq k \to \infty$. As part of these derivations we prove bounds on the mean of the error; see Remark 3.10. Our techniques can also be used to derive concentration inequalities for the $(f, \Gamma)$-divergence estimation problem, which is of independent interest for, e.g., mutual information estimation; see Theorem E.1. Due to the asymmetry of $D_f^{\Gamma}(Q \| P)$ in $Q$ and $P$, we also provide a concentration inequality where the positions of $Q$ and $P_{\theta^*_{n,m}}$ are reversed; see Theorem F.4. Finally, we note that the quantities $\mathcal{K}_{f, \widetilde{\Gamma} \circ \Phi, P_Z, m}$ and $\Delta_{f,m}$ approach their IPM counterparts in the limit where $f^*$ becomes linear; see Remarks 2.8 and 2.11.

The concentration inequality (9) and other related results presented below reduce the problem of statistical performance guarantees for $(f, \Gamma)$-GANs to the problem of bounding the Rademacher complexities of the discriminator and the composition of discriminator and generator function classes. This is a well-studied problem which also arises the case of IPM-GANs and the same techniques and results from that setting can be applied here. The discussion in Section 3.1 of Huang et al. (2022) and also Chakraborty & Bartlett (2024) shows how covering number bounds imply that, under appropriate assumptions, the IPM-GAN performance depends on the intrinsic dimension, $d^*$, of the support of $Q$ and not the (possibly much larger) ambient dimension, $d$. As our new bounds depend on the Rademacher complexity in the same qualitative manner as the IPM results, the aforementioned insights regarding dimension dependence also immediately apply to $(f, \Gamma)$-GANs. Furthermore, in Section 3.3 we discuss Rademacher complexity bounds for distributions with unbounded support. While several approaches to this problem can be found in the IPM-GAN literature, see Biau et al. (2021) and Huang et al. (2022), our result in Theorem 3.13 requires significantly less restrictive assumptions and so provides new insight even in the IPM-GAN case. Moreover, Theorem 3.13 has many applications beyond the study of GANs, such as for neural estimation of mutual information (Belghazi et al., 2018; Song & Ermon, 2020b; Birrell et al., 2022c).

## 1.2 Comparison with Generalization Bounds for the JS-GAN

The closest existing work on GANs with nonlinear objectives can be found in Arora et al. (2017), which derived statistical guarantees for nonlinear GANs with JS-style nonlinear objectives. The present work is distinguished in the following ways.

1. The nonlinearity in the $(f, \Gamma)$-GAN objective is of a different character than what was considered in Arora et al. (2017). Specifically, in addition to composing the discriminators with the nonlinear function $f^*$, the minimization over the (unbounded) shift parameter in (4) introduces several complications, as one can no longer a priori assume that $f^*$ is bounded or uniformly Lipschitz, as was assumed for $\phi$ in Arora et al. (2017) (see their Theorem 3.1); the necessity of including shifts was discussed in Birrell et al. (2022b) and also in Theorem 2 of Farnia & Tse (2018), and hence is an inherent complication of rigorously studying GANs that are based on infimal convolutions, such as $(f, \Gamma)$-GANs.

2. The statistical consistency results proven here in Theorem 3.7 and in Theorem 3.1 of Arora et al. (2017) are of different natures. The latter derives a bound on the difference between empirical and

exact errors (as measured by the chosen divergence) while our result (9) implies that the solution to the empirical GAN problem provides a near-optimal solution to the exact GAN problem. While both results are meaningful, they have different interpretations and their derivations require different techniques.

3. Finally, we note that the generalization bound in Theorem 3.1 of Arora et al. (2017) assumes the space of discriminators is Lipschitz in its parameters. Generally speaking, for discriminator architectures used in practice this property holds only when the data and noise distributions have compact support. In contrast, our novel Rademacher complexity approach in Section 3.3 allows for discriminators that satisfy a more general local Lipschitz property in their parameters. This allows our framework to be be applied to distributions with unbounded support, including a range of heavy-tailed distributions.

## 2   Properties of $\Lambda_f^P$

The fundamental difference between the more commonly studied IPM-GANs (1) and the $(f, \Gamma)$-GANs (5) is the nonlinearity of the objective functional in (3), coming from the generalized cumulant generating function (4). A detailed study of the properties of the generalized cumulant generating function, $\Lambda_f^P$, is therefore required in order to extend the techniques used for IPM-GANs to obtain statistical guarantees for $(f, \Gamma)$-GANs. We undertake that study in this section. Detailed proofs can be found in Appendix D.

Going forward, we let $\mathcal{X}$ be a measurable space and $\mathcal{M}_b(\mathcal{X})$ be the space of bounded measurable real-valued functions on $\mathcal{X}$. We let $f : (a, b) \to \mathbb{R}$, $0 \leq a < 1 < b \leq \infty$ , be a convex function that satisfies $f(1) = 0$; we denote the set of such functions by $\mathcal{F}_1(a, b)$. We will make regular use of various properties of such convex functions and their Legendre transforms that are collected in Appendix A. In particular, we note that $a \geq 0$ implies $f^*$ is non-decreasing. The value of the right derivative of $f$ at 1 will play a key role in many of the proofs and so we make the following definition:

$$z_0 := f'_+(1) \, . \tag{10}$$

By assumption, 1 is in the interior of the set where the convex function $f$ is finite and so this right derivative is guaranteed to exist and be finite.

The first key property is that bounds on a function $h$ translate to bounds on $\Lambda_f^P[h]$ and also allow the minimization over $\nu$ in (4) to be restricted to a corresponding compact interval.

**Lemma 2.1.** *Let $f \in \mathcal{F}_1(a, b)$ with $a \geq 0$, $P$ be a probability measure on $\mathcal{X}$, and $h \in \mathcal{M}_b(\mathcal{X})$ with $\alpha \leq h \leq \beta$. Then:*

1. *$\alpha \leq \Lambda_f^P[h] \leq \beta$*

2. *If $f$ is also strictly convex on a neighborhood of 1 and $z_0 \in \{f^* < \infty\}^o$ then*

$$\Lambda_f^P[h] = \inf_{\nu \in [\alpha - z_0, \beta - z_0]} \{\nu + E_P[f^*(h - \nu)]\} \, . \tag{11}$$

**Remark 2.2.** *We use $A^o$ to denote the interior of a set $A$.*

Next we give a Lipschitz bound for $\Lambda_f^P$.

**Lemma 2.3.** *Let $P$ be a probability measure on $\mathcal{X}$, and $h, \tilde{h} \in \mathcal{M}_b(\mathcal{X})$ with $\alpha \leq h, \tilde{h} \leq \beta$. Let $f \in \mathcal{F}_1(a, b)$ with $a \geq 0$ and assume $f$ is strictly convex in a neighborhood of 1 and $z_0 + \beta - \alpha \in \{f^* < \infty\}^o$. Then*

$$|\Lambda_f^P[\tilde{h}] - \Lambda_f^P[h]| \leq (f^*)'_+(z_0 + \beta - \alpha)\|\tilde{h} - h\|_{L^1(P)} \, . \tag{12}$$

We now aim to obtain a tight bound on the difference between $\Lambda_f^{P_n}[h]$ and $\Lambda_f^{\widetilde{P}_n}[h]$ when $P_n$ and $\widetilde{P}_n$ are empirical measures that differ by only a single point. This will be key for an effective use of McDiarmid's inequality, thus allowing us to derive statistical guarantees for $(f, \Gamma)$-GANs in Section 3. For this purpose it will be convenient to make the following definition.

**Definition 2.4.** *Let $f \in \mathcal{F}_1(a, b)$ with $a \geq 0$. Define $\Lambda_f : \mathbb{R}^n \to \mathbb{R}$ by*

$$\Lambda_f(x) := \inf_{\nu \in \mathbb{R}} \left\{ \nu + \frac{1}{n} \sum_{i=1}^n f^*(x_i - \nu) \right\}. \tag{13}$$

The connection to $\Lambda_f^P$ is given by the following lemma, which is a simple consequence of the respective definitions.

**Lemma 2.5.** *Let $f \in \mathcal{F}_1(a, b)$ with $a \geq 0$. Let $h \in \mathcal{M}_b(\mathcal{X})$ and $P_n = \frac{1}{n} \sum_{i=1}^n \delta_{x_i}$ be an empirical measure on $\mathcal{X}$. Then*

$$\Lambda_f^{P_n}[h] = \Lambda_f \circ h_n(x) \tag{14}$$

*where $h_n(x) := (h(x_1), ..., h(x_n))$.*

The Lipschitz bound in Lemma 2.3 translates into the following Lipschitz bound for $\Lambda_f$.

**Corollary 2.6.** *Let $f \in \mathcal{F}_1(a, b)$ with $a \geq 0$. Let $\alpha \leq \beta$ and assume $f$ is strictly convex in a neighborhood of 1 and $z_0 + \beta - \alpha \in \{f^* < \infty\}^o$. For $x, \tilde{x} \in [\alpha, \beta]^n$ we have*

$$|\Lambda_f(\tilde{x}) - \Lambda_f(x)| \leq \frac{1}{n} (f^*)'_+(z_0 + \beta - \alpha) \sum_{i=1}^n |\tilde{x}_i - x_i|. \tag{15}$$

*In particular, $\Lambda_f$ is continuous on $[\alpha, \beta]^n$.*

The following lemma gives tight perturbation bounds on $\Lambda_f$ when the inputs differ in only a single component, which in turn gives the desired tight bound on $\Lambda_f^{P_n}[h] - \Lambda_f^{\widetilde{P}_n}[h]$ when $P_n$ and $\widetilde{P}_n$ are empirical measures that differ by only a single point via Lemma 2.5.

**Lemma 2.7.** *Let $f \in \mathcal{F}_1(a, b)$ with $a \geq 0$. Let $\alpha \leq \beta$ and assume $f$ is strictly convex in a neighborhood of 1 and $z_0 + \beta - \alpha \in \{f^* < \infty\}^o$. Given $j \in \{1, ..., n\}$, let $E = \{(x, \tilde{x}) \in [\alpha, \beta]^n \times [\alpha, \beta]^n : x_i = \tilde{x}_i \text{ for } i \neq j\}$. Then*

$$\sup_{(x, \tilde{x}) \in E} \{\Lambda_f(\tilde{x}) - \Lambda_f(x)\} = \inf_{z \in [z_0 - (\beta - \alpha), z_0]} \left\{ -z + \frac{n-1}{n} f^*(z) + \frac{1}{n} f^*(\beta - \alpha + z) \right\} \tag{16}$$

*and we have the simpler loose bounds*

$$\frac{\beta - \alpha}{n} \leq \inf_{z \in [z_0 - (\beta - \alpha), z_0]} \left\{ -z + \frac{n-1}{n} f^*(z) + \frac{1}{n} f^*(\beta - \alpha + z) \right\}$$
$$\leq \frac{1}{n} (f^*(\beta - \alpha + z_0) - z_0) \leq (f^*)'_+(\beta - \alpha + z_0) \frac{\beta - \alpha}{n}. \tag{17}$$

**Remark 2.8.** *Note that (17) implies that the bound (16) continuously approaches the result in the linear case (i.e., where $\Lambda_f^P = E_P$, which corresponds to $f^*(z) = z$ on a sufficiently large interval) as $(f^*)'_+(\beta - \alpha + z_0)$ approaches 1.*

Finally, we present a pair of uniform law of large numbers (ULLN) results that bound the maximum difference between $\Lambda_f^{P_n}[h]$ and $\Lambda_f^P[h]$ over a set of test function $h \in \Gamma$ in terms of the Rademacher complexity of $\Gamma$, where $P_n$ is the empirical measure for $n$ i.i.d. samples from $P$. These results should be compared with the ULLN for means which provides uniform bounds on the difference between the empirical mean and expectation; see Appendix C for the relevant background. First we obtain a simpler result that uses (11) along with a Lipschitz bound to reduce the problem to the ULLN for means; this can be thought of as a substantially more general version of the estimate in the proof of Lemma 2 in Belghazi et al. (2018), which studied a KL-divergence based mutual-information method. However, as we will show, the resulting estimate is overly pessimistic as it does not reduce to the IPM case in the appropriate limit. Following this simpler lemma we will derive a tighter bound that does possess the desired limiting property, though it requires slightly more restrictive assumptions on $f$.

**Lemma 2.9.** *Let $\Psi$ be a nonempty countable collection of measurable functions on $\mathcal{Y}$ and suppose we have $\alpha, \beta \in \mathbb{R}$ such that $\alpha \leq \psi \leq \beta$ for all $\psi \in \Psi$. Let $f \in \mathcal{F}_1(a, b)$ with $a \geq 0$ and assume $f$ is strictly convex in a neighborhood of 1 and $z_0 + \beta - \alpha \in \{f^* < \infty\}^o$.*

*Let $P$ be a probability measure on $\mathcal{Y}$ and $Y_i$, $i = 1, ..., n$ be independent samples from $P$ with $P_n$ the corresponding empirical measure. Then*

$$\mathbb{E}\left[\sup_{\psi \in \Psi}\left\{\pm\left(\Lambda_f^P[\psi] - \Lambda_f^{P_n}[\psi]\right)\right\}\right] \leq 2(f^*)'_+(\beta - \alpha + z_0)\left(\mathcal{R}_{\Psi,P,n} + \frac{\beta - \alpha}{2n^{1/2}}\right). \tag{18}$$

**Lemma 2.10.** *Let $\Psi$ be a countable collection of measurable functions on $\mathcal{Y}$ that contains at least one constant function and suppose we have $\alpha, \beta \in \mathbb{R}$ such that $\alpha \leq \psi \leq \beta$ for all $\psi \in \Psi$. Let $f \in \mathcal{F}_1(a, b)$ with $a \geq 0$ and assume $f$ is strictly convex in a neighborhood of 1 and $z_0 + \beta - \alpha \in \{f^* < \infty\}^o$. Also assume that $(f^*)'_+$ is $L_{\alpha,\beta}$-Lipschitz on $[z_0 - (\beta - \alpha), z_0 + \beta - \alpha]$.*

*Let $P$ be a probability measure on $\mathcal{Y}$ and $Y_i$, $i = 1, ..., n$ be independent samples from $P$ with $P_n$ the corresponding empirical measure. Then*

$$\mathbb{E}\left[\sup_{\psi \in \Psi}\left\{\pm\left(\Lambda_f^P[\psi] - \Lambda_f^{P_n}[\psi]\right)\right\}\right] \tag{19}$$

$$\leq 2\min\left\{(1 + 2(\beta - \alpha)L_{\alpha,\beta})\mathcal{R}_{\Psi,P,n} + \frac{(\beta - \alpha)^2 L_{\alpha,\beta}}{2n^{1/2}}, (f^*)'_+(\beta - \alpha + z_0)\left(\mathcal{R}_{\Psi,P,n} + \frac{\beta - \alpha}{2n^{1/2}}\right)\right\}.$$

**Remark 2.11.** *The result in Lemma 2.9 does not reduce to the ULLN for means in the case where $f^*(z) = z$ (on a sufficiently large interval), i.e., when $\Lambda_f^P[h] = E_P[h]$; specifically the bound (18) differs from the ULLN for means, as recalled in (64) of Appendix C, by the term $\frac{\beta - \alpha}{m^{1/2}}$ in that case. However, (19) does reduce to the ULLN for means when $f^*(z) = z$. Moreover, the bound (19) approaches the bound in (64) as the Lipschitz constant for $(f^*)'_+$ on $[z_0 - (\beta - \alpha), z_0 + \beta - \alpha]$ approaches zero. We emphasize that, when comparing our results with the IPM case, the sub-optimal second term in (2.9), which comes from the parameter $\nu$, cannot be absorbed into the Rademacher complexity as the bias of the final layer. This is because the bias of the final layer always cancels in an IPM and hence does not contribute to the Rademacher complexity terms for IPM-GANs. Thus the improved Lemma 2.10 is necessary in order to derive a general result that appropriately reduces to the IPM case.*

## 3 Error Bounds for $(f, \Gamma)$-GANs

In this section we use the properties of $\Lambda_f^P$ from Section 2 to derive concentration inequalities for $(f, \Gamma)$-GANs.

### 3.1 $(f, \Gamma)$-GAN Error Decomposition

We start by deriving a decomposition of the $(f, \Gamma)$-GAN error into statistical, approximation, and optimization error terms. First we consider the error that comes from approximating the idealized discriminator space $\Gamma$ by a smaller space (e.g., a neural network) $\widetilde{\Gamma}$, a step that is generally required when implementing a GAN method:

**Lemma 3.1.** *Let $\widetilde{\Gamma} \subset \Gamma \subset \mathcal{M}_b(\mathcal{X})$ be nonempty and suppose we have $\alpha, \beta \in \mathbb{R}$ such that $\alpha \leq h \leq \beta$ for all $h \in \Gamma$. Let $f \in \mathcal{F}_1(a, b)$ with $a \geq 0$ and assume $f$ is strictly convex in a neighborhood of 1 and $z_0 + \beta - \alpha \in \{f^* < \infty\}^o$. Then for any probability measures $Q, P$ on $\mathcal{X}$ we have*

$$0 \leq D_f^\Gamma(Q\|P) - D_f^{\widetilde{\Gamma}}(Q\|P) \leq \left(1 + (f^*)'_+(z_0 + \beta - \alpha)\right)\sup_{h \in \Gamma}\inf_{\tilde{h} \in \widetilde{\Gamma}}\|h - \tilde{h}\|_\infty. \tag{20}$$

*Proof.* For any $h \in \Gamma$, $\tilde{h} \in \widetilde{\Gamma}$ we can use the definition (3) and Lemma 2.3 to compute

$$
\begin{aligned}
E_Q[h] - \Lambda_f^P[h] - D_f^{\widetilde{\Gamma}}(Q\|P) &\leq E_Q[h] - E_Q[\tilde{h}] + \Lambda_f^P[\tilde{h}] - \Lambda_f^P[h] \\
&\leq \|h - \tilde{h}\|_{L^1(Q)} + (f^*)'_+(z_0 + \beta - \alpha)\|\tilde{h} - h\|_{L^1(P)} \\
&\leq \left(1 + (f^*)'_+(z_0 + \beta - \alpha)\right)\|h - \tilde{h}\|_\infty .
\end{aligned}
\tag{21}
$$

Minimizing over $\tilde{h} \in \widetilde{\Gamma}$ gives

$$
E_Q[h] - \Lambda_f^P[h] - D_f^{\widetilde{\Gamma}}(Q\|P) \leq \left(1 + (f^*)'_+(z_0 + \beta - \alpha)\right) \inf_{\tilde{h} \in \widetilde{\Gamma}} \|h - \tilde{h}\|_\infty
\tag{22}
$$

for all $h \in \Gamma$. Maximizing over $h \in \Gamma$ then completes the proof. $\qquad\square$

Next we outline the assumptions we make regarding the discriminator, generator, and the empirical GAN optimization.

**Assumption 3.2** (($f, \Gamma$)-GAN Assumptions). *Let $f \in \mathcal{F}_1(a, b)$ with $a \geq 0$ and $\alpha, \beta \in \mathbb{R}$, $\alpha < \beta$ that satisfy the following:*

1. *$f$ is strictly convex in a nbhd of $1$.*

2. *$z_0 + \beta - \alpha \in \{f^* < \infty\}^o$, where $z_0$ was defined in (10).*

*Let $(\mathcal{X}, \mathcal{B}_\mathcal{X})$ be a topological space with the Borel sigma algebra. Suppose $\widetilde{\Gamma} \subset \Gamma \subset C_b(\mathcal{X})$ (the space of bounded continuous functions) are nonempty (the discriminator spaces) and satisfy the following:*

1. *$\alpha \leq h \leq \beta$ for all $h \in \Gamma$.*

2. *There exists a countable $\Gamma_0 \subset \Gamma$ such that for all $h \in \Gamma$ there exists a sequence $h_j \in \Gamma_0$ such that $h_j \to h$ pointwise.*

3. *There exists a countable $\widetilde{\Gamma}_0 \subset \widetilde{\Gamma}$ such that for all $\tilde{h} \in \widetilde{\Gamma}$ there exists a sequence $\tilde{h}_j \in \widetilde{\Gamma}_0$ such that $\tilde{h}_j \to \tilde{h}$ pointwise.*

*Let $\mathcal{Z}$ be another measurable space, $P_Z$ a probability measure on $\mathcal{Z}$, and $\Phi_\theta : \mathcal{Z} \to \mathcal{X}$ be measurable for $\theta \in \Theta$, where $\Theta$ is a separable metric space. Suppose $\theta \mapsto \Phi_\theta(z)$ is continuous for all $z \in \mathcal{Z}$. Define $P_\theta := (\Phi_\theta)_\# P_Z$, which is a probability measure on $\mathcal{X}$.*

*Let $Q$ be a probability measure on $\mathcal{X}$ and $X_i$, $i = 1, ..., n$, $Z_i$, $i = 1, ..., m$ be independent and distributed as $Q$, $P_Z$ respectively. Let $Q_n$, $P_{Z,m}$. and $P_{\theta,m}$ be the empirical measures corresponding to $X_i$, $Z_i$, and $\Phi_\theta \circ Z_i$ respectively. Finally, suppose that for each $m, n$ we have an error tolerance $\epsilon_{n,m}^{opt} \geq 0$ and $\Theta$-valued random variables $\theta_{n,m}^*$ that are approximate optimizers to the empirical GAN problem, i.e., that satisfy*

$$
D_f^{\widetilde{\Gamma}}(Q_n\|P_{\theta_{n,m}^*,m}) \leq \inf_{\theta \in \Theta} D_f^{\widetilde{\Gamma}}(Q_n\|P_{\theta,m}) + \epsilon_{n,m}^{opt} \quad \mathbb{P}\text{-}a.s.
\tag{23}
$$

**Remark 3.3.** *The assumptions regarding $\Gamma_0$, $\widetilde{\Gamma}_0$, separability of $\Theta$, and continuity of $\Phi_\theta$ and $h \in \Gamma$ allow us to address the issue of measurability of the various suprema that arise in the derivations below by enabling one to restrict them to countable subsets. These assumptions hold in most cases of interest and can also be replaced by any alternatives that serve the same purpose.*

Several important examples of $f$'s that satisfy the required assumptions can be found in Table 1. Specifically, these examples all satisfy $z_0 \in \{f^* < \infty\}^o$ and hence $z_0 + \beta - \alpha \in \{f^* < \infty\}^o$ for appropriate choices of $\alpha, \beta$.

The concentration inequalities that we derive below rely on the following decompositions of the ($f, \Gamma$)-GAN error. These should be compared with the result for IPM-GANs, see Lemma 9 of Huang et al. (2022).

| Divergence | $f(t)$ | $f^*(z)$ | $z_0$ |
|---|---|---|---|
| JS-divergence | $t\log(t) - (t+1)\log\left(\frac{1+t}{2}\right)$ | $-\log(2 - e^z)$ | $0$ |
| KL-divergence | $t\log(t)$ | $e^{z-1}$ | $1$ |
| $\alpha$-divergence ($\alpha > 1$) | $\frac{t^\alpha - 1}{\alpha(\alpha-1)}$ | $\alpha^{-1}(\alpha-1)^{\alpha/(\alpha-1)}\max\{z,0\}^{\alpha/(\alpha-1)} + \frac{1}{\alpha(\alpha-1)}$ | $\frac{1}{\alpha-1}$ |

Table 1: Convex functions, $f$, that satisfy the assumptions required by the theorems below, along with their convex conjugates, $f^*$, and the value of $z_0$, as defined in (10). Note that the representation of the Jensen-Shannon divergence obtained by combining the first row of the table with (6) is different from the one used in the original GAN (Goodfellow et al., 2014) or in the analysis of JS-style GANs in Arora et al. (2017).

**Lemma 3.4** (($f,\Gamma$)-GAN Error Decomposition). *Under the Assumption 3.2 the ($f,\Gamma$)-GAN error can be decomposed* $\mathbb{P}$-*a.s. as follows:*

$$D_f^\Gamma(Q\|P_{\theta_{n,m}^*}) - \inf_{\theta\in\Theta} D_f^\Gamma(Q\|P_\theta) \tag{24}$$

$$\leq \sup_{h\in\widetilde{\Gamma},\theta\in\Theta}\left\{\Lambda_f^{P_{Z,m}}[h\circ\Phi_\theta] - \Lambda_f^{P_Z}[h\circ\Phi_\theta])\right\} + \sup_{h\in\widetilde{\Gamma},\theta\in\Theta}\left\{\Lambda_f^{P_Z}[h\circ\Phi_\theta] - \Lambda_f^{P_{Z,m}}[h\circ\Phi_\theta]\right\}$$

$$+ \sup_{h\in\widetilde{\Gamma}}\left\{E_Q[h] - E_{Q_n}[h]\right\} + \sup_{h\in\widetilde{\Gamma}}\left\{E_{Q_n}[h] - E_Q[h]\right\}$$

$$+ \left(1 + (f^*)'_+(z_0 + \beta - \alpha)\right)\sup_{h\in\Gamma}\inf_{\tilde{h}\in\widetilde{\Gamma}}\|h - \tilde{h}\|_\infty + \epsilon_{n,m}^{opt}.$$

*Proof.* Using (23) we can compute the $\mathbb{P}$-a.s. bound

$$D_f^\Gamma(Q\|P_{\theta_{n,m}^*}) - \inf_{\theta\in\Theta} D_f^\Gamma(Q\|P_\theta) \tag{25}$$

$$= D_f^\Gamma(Q\|P_{\theta_{n,m}^*}) - \inf_{\theta\in\Theta} D_f^{\widetilde{\Gamma}}(Q_n\|P_{\theta,m}) + \inf_{\theta\in\Theta} D_f^{\widetilde{\Gamma}}(Q_n\|P_{\theta,m}) - \inf_{\theta\in\Theta} D_f^\Gamma(Q\|P_\theta)$$

$$\leq D_f^\Gamma(Q\|P_{\theta_{n,m}^*}) - D_f^{\widetilde{\Gamma}}(Q_n\|P_{\theta_{n,m}^*,m}) + \epsilon_{n,m}^{opt} + \inf_{\theta\in\Theta} D_f^{\widetilde{\Gamma}}(Q_n\|P_{\theta,m}) - \inf_{\theta\in\Theta} D_f^\Gamma(Q\|P_\theta).$$

Using the fact that $\widetilde{\Gamma} \subset \Gamma$ implies $D_f^{\widetilde{\Gamma}} \leq D_f^\Gamma$ along with Lemma 3.1 we then find

$$D_f^\Gamma(Q\|P_{\theta_{n,m}^*}) - \inf_{\theta\in\Theta} D_f^\Gamma(Q\|P_\theta) \tag{26}$$

$$\leq D_f^{\widetilde{\Gamma}}(Q\|P_{\theta_{n,m}^*}) - D_f^{\widetilde{\Gamma}}(Q_n\|P_{\theta_{n,m}^*,m}) + \inf_{\theta\in\Theta} D_f^{\widetilde{\Gamma}}(Q_n\|P_{\theta,m}) - \inf_{\theta\in\Theta} D_f^{\widetilde{\Gamma}}(Q\|P_\theta)$$

$$+ \left(1 + (f^*)'_+(z_0 + \beta - \alpha)\right)\sup_{h\in\Gamma}\inf_{\tilde{h}\in\widetilde{\Gamma}}\|h - \tilde{h}\|_\infty + \epsilon_{n,m}^{opt}.$$

Next we make use of the simple bound

$$\pm\left(\sup_{i\in I} d_i - \sup_{i\in I} c_i\right) \leq \sup_{i\in I}\{\pm(d_i - c_i)\} \tag{27}$$

whenever $c_i, d_i \in \mathbb{R}$ for all $i \in I$ and $\sup_i c_i, \sup_i d_i$ are finite. Using the definition (3) along with (78) and (27) we can compute

$$
\begin{aligned}
D_f^\Gamma(Q\|P_{\theta_{n,m}^*}) - \inf_{\theta\in\Theta} D_f^\Gamma(Q\|P_\theta) &\leq \sup_{h\in\widetilde\Gamma}\left\{ E_Q[h] - \Lambda_f^{P_{\theta_{n,m}^*}}[h] - \left( E_{Q_n}[h] - \Lambda_f^{P_{\theta_{n,m}^*},m}[h]\right)\right\} \qquad (28)\\
&\quad + \sup_{\theta\in\Theta}\sup_{h\in\widetilde\Gamma}\left\{ E_{Q_n}[h] - \Lambda_f^{P_\theta,m}[h] - \left(E_Q[h] - \Lambda_f^{P_\theta}[h]\right)\right\}\\
&\quad + \left(1 + (f^*)'_+(z_0+\beta-\alpha)\right)\sup_{h\in\Gamma}\inf_{\tilde h\in\widetilde\Gamma}\|h-\tilde h\|_\infty + \epsilon_{n,m}^{opt}\\
&\leq \sup_{h\in\widetilde\Gamma}\left\{ E_Q[h] - E_{Q_n}[h]\right\} + \sup_{h\in\widetilde\Gamma, \theta\in\Theta}\left\{\Lambda_f^{P_\theta,m}[h] - \Lambda_f^{P_\theta}[h]\right\}\\
&\quad + \sup_{h\in\widetilde\Gamma}\left\{ E_{Q_n}[h] - E_Q[h]\right\} + \sup_{h\in\widetilde\Gamma, \theta\in\Theta}\left\{\Lambda_f^{P_\theta}[h] - \Lambda_f^{P_\theta,m}[h]\right\}\\
&\quad + \left(1 + (f^*)'_+(z_0+\beta-\alpha)\right)\sup_{h\in\Gamma}\inf_{\tilde h\in\widetilde\Gamma}\|h-\tilde h\|_\infty + \epsilon_{n,m}^{opt}.
\end{aligned}
$$

We have $\Lambda_f^{P_\theta}[h] = \Lambda_f^{P_Z}[h\circ\Phi_\theta]$ and $\Lambda_f^{P_\theta,m}[h] = \Lambda_f^{P_Z,m}[h\circ\Phi_\theta]$ and so this completes the proof. $\qquad\square$

We note that the setting of Lemmas 3.4 differs in a few ways from that of Huang et al. (2022). Namely we assume $\widetilde\Gamma\subset\Gamma$ and we do not assume that $\inf_{\theta\in\Theta} D_f^\Gamma(Q\|P_\theta) = 0$. Under appropriate assumptions, the latter can be proven to hold for a sufficiently rich class of generators. More specifically, the analogue of the approach in Huang et al. (2022) would be to work under the assumption that $\inf_{\theta\in\Theta} D_f^{\widetilde\Gamma}(\mu_n\|P_\theta) = 0$ for all empirical distributions $\mu_n$. As $D_f^{\widetilde\Gamma} \leq d_{\widetilde\Gamma}$ (see (60) in Appendix B), this zero generator approximation error property holds for the $(f,\Gamma)$-divergence whenever it holds for the the corresponding IPM; see Yang et al. (2022) and the discussion in Section 2.2.1 of Huang et al. (2022) for sufficient conditions. Below we give an error decompositions that is adapted to the zero-approximation-error; the proof, which is very similar to that of Lemma 3.4, can be found in Appendix D.

**Lemma 3.5** (($f,\Gamma$)-GAN Error Decomposition 2). *Under Assumption 3.2, and supposing that $\inf_{\theta\in\Theta} D_f^{\widetilde\Gamma}(\mu_n\|P_\theta) = 0$ for all empirical distributions $\mu_n$, the $(f,\Gamma)$-GAN error can be decomposed $\mathbb{P}$-a.s. as follows:*

$$
\begin{aligned}
D_f^\Gamma(Q\|P_{\theta_{n,m}^*}) &\leq \sup_{h\in\widetilde\Gamma,\theta\in\Theta}\left\{\Lambda_f^{P_Z,m}[h\circ\Phi_\theta] - \Lambda_f^{P_Z}[h\circ\Phi_\theta]\right\} + \sup_{h\in\widetilde\Gamma,\theta\in\Theta}\left\{\Lambda_f^{P_Z}[h\circ\Phi_\theta] - \Lambda_f^{P_Z,m}[h\circ\Phi_\theta]\right\} \qquad (29)\\
&\quad + \sup_{h\in\widetilde\Gamma}\left\{ E_Q[h] - E_{Q_n}[h]\right\} + \left(1 + (f^*)'_+(z_0+\beta-\alpha)\right)\sup_{h\in\Gamma}\inf_{\tilde h\in\widetilde\Gamma}\|h-\tilde h\|_\infty + \epsilon_{n,m}^{opt}.
\end{aligned}
$$

Note that the terms in the bounds (29) reduce to the terms in IPM case, Lemma 9 in Huang et al. (2022), if $f^*(z) = z$ and $-\widetilde\Gamma\subset\widetilde\Gamma$.

## 3.2 Concentration Inequalities for $(f,\Gamma)$-GANs

We are now ready to derive concentration inequalities for $(f,\Gamma)$-GANs; we present two variants, depending on whether one assumes the zero generator approximation error property for empirical measures. The key ingredients are the ULLN for the generalized cumulant generating function in Lemmas 2.9 and 2.10 along with the perturbation bound from Lemma 2.7, the latter being needed in order to apply McDiarmid's inequality to the terms involving the generalized cumulant generating function. The following definition lists several quantities that will appear in the concentration inequalities below. We note that these quantities approach their counterparts in the linear (i.e., IPM) case in the appropriate limit; see the discussion in Remarks 2.8 and 2.11.

**Definition 3.6.** *With notation as in either Assumption 3.2 or F.1:*

1. *Define the discriminator-space approximation error*

$$\epsilon_{approx}^{\Gamma,\widetilde{\Gamma}} := \left(1 + (f^*)'_+(z_0 + \beta - \alpha)\right) \sup_{h \in \Gamma} \inf_{\tilde{h} \in \widetilde{\Gamma}} \|h - \tilde{h}\|_\infty . \tag{30}$$

*Note that this vanishes when $\widetilde{\Gamma} = \Gamma$.*

2. *For $n \in \mathbb{Z}^+$ define*

$$\Delta_{f,n} := \inf_{z \in [z_0 - (\beta - \alpha), z_0]} \left\{ n \left( -z + \frac{n-1}{n} f^*(z) + \frac{1}{n} f^*(\beta - \alpha + z) \right) \right\} \tag{31}$$

*and note that $\Delta_{f,n} \in [\beta - \alpha, f^*(\beta - \alpha + z_0) - z_0]$.*

3. *Let $n \in \mathbb{Z}^+$, $\Psi$ be a nonempty family of measurable functions on $\mathcal{X}$, and $P$ a probability measure on $\mathcal{X}$. If $(f^*)'_+$ is $L_{\alpha,\beta}$-Lipschitz on $[z_0 - (\beta - \alpha), z_0 + \beta - \alpha]$ and $\Psi$ contains a constant function then define*

$$\mathcal{K}_{f,\Psi,P,n} \tag{32}$$
$$:= \min \left\{ (1 + 2(\beta - \alpha) L_{\alpha,\beta}) \mathcal{R}_{\Psi,P,n} + \frac{(\beta - \alpha)^2 L_{\alpha,\beta}}{2n^{1/2}}, (f^*)'_+(\beta - \alpha + z_0) \left( \mathcal{R}_{\Psi,P,n} + \frac{\beta - \alpha}{2n^{1/2}} \right) \right\}$$

*and otherwise define*

$$\mathcal{K}_{f,\Psi,P,n} := (f^*)'_+(\beta - \alpha + z_0) \left( \mathcal{R}_{\Psi,P,n} + \frac{\beta - \alpha}{2n^{1/2}} \right) . \tag{33}$$

With these definitions, we present the following concentration inequalities.

**Theorem 3.7** (($f, \Gamma$)-GAN Concentration Inequalities). *Under Assumption 3.2, and in particular with $\theta_{n,m}^*$ the approximate solution to the empirical $(f, \Gamma)$-GAN problem (23), for $\epsilon > 0$ we have*

$$\mathbb{P} \left( D_f^\Gamma(Q \| P_{\theta_{n,m}^*}) - \inf_{\theta \in \Theta} D_f^\Gamma(Q \| P_\theta) \geq \epsilon + \epsilon_{approx}^{\Gamma,\widetilde{\Gamma}} + \epsilon_{opt}^{n,m} + 4\mathcal{R}_{\widetilde{\Gamma},Q,n} + 4\mathcal{K}_{f,\widetilde{\Gamma} \circ \Phi, P_Z, m} \right) \tag{34}$$

$$\leq \exp \left( -\frac{\epsilon^2}{\frac{2}{n}(\beta - \alpha)^2 + \frac{2}{m} \Delta_{f,m}^2} \right) ,$$

*where we refer to the quantities in Definition 3.6.*

*If, in addition, $\inf_{\theta \in \Theta} D_f^{\widetilde{\Gamma}}(\mu_n \| P_\theta) = 0$ for all possible empirical distributions $\mu_n$ then we obtain the tighter bound*

$$\mathbb{P} \left( D_f^\Gamma(Q \| P_{\theta_{n,m}^*}) \geq \epsilon + \epsilon_{approx}^{\Gamma,\widetilde{\Gamma}} + \epsilon_{opt}^{n,m} + 2\mathcal{R}_{\widetilde{\Gamma},Q,n} + 4\mathcal{K}_{f,\widetilde{\Gamma} \circ \Phi, P_Z, m} \right) \leq \exp \left( -\frac{\epsilon^2}{\frac{1}{2n}(\beta - \alpha)^2 + \frac{2}{m} \Delta_{f,m}^2} \right) . \tag{35}$$

**Remark 3.8.** *See Table 1 for several important examples of nonlinear $f$'s to which this theorem applies.*

*Proof.* First note that Lemma 2.3 and Corollary 2.6 together with the dominated convergence theorem imply $\theta \mapsto \Lambda_f^{P_Z}[h \circ \Phi_\theta]$ is are continuous and also that $h_j \to h$ pointwise implies $\Lambda_f^{P_Z}[h_j \circ \Phi_\theta] \to \Lambda_f^{P_Z}[h \circ \Phi_\theta]$, $E_Q[h_j] \to E_Q[h]$, and similarly for the empirical variants. Therefore, letting $\Theta_0$ denote a countable dense subset of $\Theta$, the suprema in Lemma 3.4 can be restricted to the countable subsets $\widetilde{\Gamma}_0$ and $\Theta_0$, giving the $\mathbb{P}$-a.s. bound

$$D_f^\Gamma(Q \| P_{\theta_{n,m}^*}) - \inf_{\theta \in \Theta} D_f^\Gamma(Q \| P_\theta) \tag{36}$$

$$\leq \sup_{h \in \widetilde{\Gamma}_0, \theta \in \Theta_0} \left\{ \Lambda_f^{P_Z,m}[h \circ \Phi_\theta] - \Lambda_f^{P_Z}[h \circ \Phi_\theta]) \right\} + \sup_{h \in \widetilde{\Gamma}_0, \theta \in \Theta_0} \left\{ \Lambda_f^{P_Z}[h \circ \Phi_\theta] - \Lambda_f^{P_Z,m}[h \circ \Phi_\theta] \right\}$$

$$+ \sup_{h \in \widetilde{\Gamma}_0} \{ E_Q[h] - E_{Q_n}[h] \} + \sup_{h \in \widetilde{\Gamma}_0} \{ E_{Q_n}[h] - E_Q[h] \}$$

$$+ \epsilon_{approx}^{\Gamma,\widetilde{\Gamma}} + \epsilon_{n,m}^{opt} .$$

Now we will apply McDiarmid's inequality to the right-hand side. The map $H : \mathcal{X}^n \times \mathcal{Z}^m \to \mathbb{R}$ defined by

$$H(x,z) = \sup_{h \in \widetilde{\Gamma}_0, \theta \in \Theta_0} \left\{ \Lambda_f^{P_{Z,m}}[h \circ \Phi_\theta] - \Lambda_f^{P_Z}[h \circ \Phi_\theta]) \right\} + \sup_{h \in \widetilde{\Gamma}_0, \theta \in \Theta_0} \left\{ \Lambda_f^{P_Z}[h \circ \Phi_\theta] - \Lambda_f^{P_{Z,m}}[h \circ \Phi_\theta] \right\} \quad (37)$$
$$+ \sup_{h \in \widetilde{\Gamma}_0} \left\{ E_Q[h] - E_{Q_n}[h] \right\} + \sup_{h \in \widetilde{\Gamma}_0} \left\{ E_{Q_n}[h] - E_Q[h] \right\}$$

is measurable and if $x, \tilde{x} \in \mathcal{X}^n$ differ only in the $j$'th component then

$$|H(x,z) - H(\tilde{x},z)| \leq 2 \sup_{h \in \widetilde{\Gamma}_0} \left\{ \left| \frac{1}{n} \sum_{i=1}^n h(x_i) - \frac{1}{n} \sum_{i=1}^n h(\tilde{x}_i) \right| \right\} \quad (38)$$
$$\leq \frac{2}{n} \sup_{h \in \widetilde{\Gamma}_0} \left\{ |h(x_j) - h(x_j')| \right\} \leq \frac{2}{n}(\beta - \alpha),$$

while if $z, \tilde{z} \in \mathcal{Z}^m$ differ only in the $j$'th component then Lemma 2.5 and the perturbation bound from Lemma 2.7 imply

$$|H(x,z) - H(x,\tilde{z})| \leq 2 \sup_{h \in \widetilde{\Gamma}_0, \theta \in \Theta_0} \left\{ |\Lambda_f \circ (h \circ \Phi_\theta)_m(z) - \Lambda_f \circ (h \circ \Phi_\theta)_m(\tilde{z})| \right\} \leq \frac{2}{m} \Delta_{f,m}. \quad (39)$$

Therefore we can apply McDiarmid's inequality to $H$, see, e.g., Theorem D.8 in Mohri et al. (2018), to obtain

$$\mathbb{P}\left( H(X,Z) \geq \epsilon + \mathbb{E}[H(X,Z)] \right) \leq \exp\left( -\frac{\epsilon^2}{\frac{2}{n}(\beta - \alpha)^2 + \frac{2}{m}\Delta_{f,m}^2} \right). \quad (40)$$

In terms of $H$, the error bound (36) becomes the a.s. bound

$$D_f^\Gamma(Q\|P_{\theta_{n,m}^*}) - \inf_{\theta \in \Theta} D_f^\Gamma(Q\|P_\theta) \leq H(X,Z) + \epsilon_{approx}^{\Gamma, \widetilde{\Gamma}} + \epsilon_{n,m}^{opt}. \quad (41)$$

Using the ULLN for means, see Theorem C.4, along with the new ULLN for the generalized cumulant generating function in Lemmas 2.9 and 2.10 we obtain

$$\mathbb{E}[H(X,Z)] \leq 4\mathcal{R}_{\widetilde{\Gamma},Q,n} + 4\mathcal{K}_{f,\widetilde{\Gamma}\circ\Phi,P_Z,m}. \quad (42)$$

Combining (40), (41), and (42) we arrive at the claimed result (34).

If we also assume $\inf_{\theta \in \Theta} D_f^{\widetilde{\Gamma}}(\mu_n\|P_\theta) = 0$ for all possible empirical distributions $\mu_n$ then we can write the error decomposition 3.5 as

$$D_f^\Gamma(Q\|P_{\theta_{n,m}^*}) \leq H(X,Z) + \epsilon_{approx}^{\Gamma, \widetilde{\Gamma}} + \epsilon_{n,m}^{opt} \quad (43)$$

$\mathbb{P}$-a.s., where we now define

$$H(X,Z) := \sup_{h \in \widetilde{\Gamma}_0, \theta \in \Theta_0} \left\{ \Lambda_f^{P_{Z,m}}[h \circ \Phi_\theta] - \Lambda_f^{P_Z}[h \circ \Phi_\theta] \right\} + \sup_{h \in \widetilde{\Gamma}_0, \theta \in \Theta_0} \left\{ \Lambda_f^{P_Z}[h \circ \Phi_\theta] - \Lambda_f^{P_{Z,m}}[h \circ \Phi_\theta] \right\} \quad (44)$$
$$+ \sup_{h \in \widetilde{\Gamma}_0} \left\{ E_Q[h] - E_{Q_n}[h] \right\}.$$

Similar to the above, when $x$ and $\tilde{x}$ differ in only a single component we can bound

$$|H(x,z) - H(\tilde{x},z)| \leq \frac{1}{n}(\beta - \alpha) \quad (45)$$

and when $z$ and $\tilde{z}$ differ in only a single component we can bound

$$|H(x,z) - H(x,\tilde{z})| \leq \frac{2}{m} \Delta_{f,m}. \quad (46)$$

The mean of $H(X,Z)$ can be bounded via Theorem C.4 and Lemmas 2.9 and 2.10:

$$\mathbb{E}[H(X,Z)] \leq 2\mathcal{R}_{\widetilde{\Gamma},Q,n} + 4\mathcal{K}_{f,\widetilde{\Gamma}\circ\Phi,P_Z,m}. \quad (47)$$

Combining (43), and (45) - (47) with McDiarmid's inequality completes the proof of (35). □

We also note that the methods employed above can also be used to obtain concentration inequalities for the estimation of $D_f^\Gamma(Q\|P)$ from samples; see Appendix E for details. Due to the asymmetry of $D_f^\Gamma$ in its arguments, it also desirable to obtain concentration inequalities for the reverse $(f,\Gamma)$-GANs, i.e., with the role of the data source and generator reversed. The analysis in this case is very similar and the corresponding results can be found in Appendix F.

By combining Remarks 2.8 and 2.11 one finds that the result of Theorem 3.7 reduces to the IPM-GAN case when $f^*$ is linear (on a sufficiently large interval). Moreover, the terms appearing in the bound continuously approach their IPM counterparts as $f^*(z)$ approaches $z$ in the appropriate sense, as outlined in the aforementioned remarks. Moreover, in practice, while $n$ is often inherently constrained by the availability of data, there is no inherent constraint on $m$, as $P_Z$ is specifically chosen so that it is easy to sample from. Therefore one can choose to work with $m \gg n$, in which case the terms involving $\frac{1}{m}\Delta_{f,m}^2$ on the right-hand sides of (34) and (35) are negligible. Similarly, under appropriate assumptions, $\mathcal{K}_{f,\widetilde{\Gamma}\circ\Phi,P_Z,m}$ will be negligible compared to $R_{\widetilde{\Gamma},Q,n}$; see Theorem 3.13. Therefore, apart from the approximation and optimization errors, the error terms in (34) and (35) will be nearly the same as those in the IPM case in the regime $m \gg n$. The same is not true if one reverses the order of $Q$ and $P_\theta$ in the arguments of the divergence, as in Theorem F.4.

**Remark 3.9.** *As stated, the above results assume a space of discriminators, $\Gamma$, that satisfies a uniform bound of the form $\alpha \le h \le \beta$ for all $h \in \Gamma$. However, we note that the $(f,\Gamma)$-divergence objective functional in (3) is invariant under constant shifts, due to the identity $\Lambda_f^P[h+c] = \Lambda_f^P[h] + c$ for all $c \in \mathbb{R}$. Therefore, if we have discriminators $\Psi$ such that the ranges of $\psi \in \Psi$ have uniformly bounded diameter, i.e., there exists $\beta$ such that $\sup_{x,\tilde{x}} |\psi(x) - \psi(\tilde{x})| \le \beta$ for all $\psi \in \Psi$, then we can write*

$$D_f^\Psi(Q\|P) = D_f^\Gamma(Q\|P)\,, \tag{48}$$

*where $\Gamma := \{\psi - \inf \psi : \psi \in \Psi\}$ satisfies $0 \le h \le \beta$ for all $h \in \Gamma$. Thus our theory can be applied to $D_f^\Gamma$ and hence also to $D_f^\Psi$ via (48). In this way, our theorems can be applied to $(f,\Gamma)$-GANs with discriminators whose ranges have uniformly bounded diameter, e.g., 1-Lipschitz functions on a compact domain.*

**Remark 3.10** ($L^q$-Bounds on the $(f,\Gamma)$-GAN error). *Bounds on the mean of the $(f,\Gamma)$-GAN error are implicit in the above derivations. They are obtained by combining the error decompositions in Section 3.1 with the ULLN results in Lemma 2.9, Lemma 2.10, and Theorem C.4. In addition, a standard technique can be used to turn the concentration inequalities into $L^q$ error bounds for any $q > 1$ as follows: Let $Y$ be a non-negative random variable, $a \in [0,\infty)$, and $K : (0,\infty) \to [0,\infty)$ be measurable such that*

$$\mathbb{P}(Y \ge \epsilon + a) \le K(\epsilon) \tag{49}$$

*for all $\epsilon > 0$. Then for $q > 0$ we have*

$$\mathbb{E}[Y^q] \le a^q + q \int_0^\infty (a+\epsilon)^{q-1} K(\epsilon) d\epsilon\,. \tag{50}$$

*This follows from rewriting*

$$\mathbb{E}|Y^q| = \int_0^\infty \mathbb{P}(Y^q \ge r) dr\,, \tag{51}$$

*breaking the domain of integration into $[0,a^q]$ and $[a^q,\infty)$, then changing variables in the second term and using the bound (49).*

## 3.3 Rademacher Complexity Bounds for Distributions with Unbounded Support

The $(f,\Gamma)$-GAN concentration inequalities derived above do not explicitly make any assumptions regarding the distributions $Q$ or $P_Z$. However, for the bounds to be meaningful one requires additional assumptions to ensure the Rademacher complexities are finite and approach zero as the number of samples increases to infinity. If the distributions have compact support, the decay of the Rademacher complexity for typical discriminator and generator classes (e.g., neural networks) follows from standard covering number arguments

without any further assumptions on the distributions. However, the case of unbounded support is more subtle. As the the noise source $P_Z$ is often chosen to be Gaussian in practice, the ability to handle distributions with unbounded support is of interest even when the data distribution naturally has compact support. In this section we provide an approach to this problem that only assumes the distributions have finite second moments, as opposed to the much stronger assumptions made in previous approaches, e.g., the sub-Gaussian and sub-exponential assumptions required in Proposition 20 of Biau et al. (2021) and Theorem 22 of Huang et al. (2022) respectively.

**Assumption 3.11.** *Suppose $(\Theta, d_\Theta)$ is a metric space and we have a collection of measurable real-valued functions on $\mathcal{Y}$, $\Psi = \{\psi_\theta : \theta \in \Theta\}$ that satisfy the following properties.*

    *1. For all $y \in \mathcal{Y}$ there exists $L(y) > 0$ such that $\theta \mapsto \psi_\theta(y)$ is $L(y)$-Lipschitz.*

    *2. $P$ is a probability measure on $\mathcal{Y}$ and $L \in L^2(P)$.*

**Remark 3.12.** *In this section we are thinking of the discriminator as being a parameterized family of functions (e.g., a NN) and $\psi_\theta$, $\theta \in \Theta$ represents either the discriminator or the composition of discriminator and generator, depending on which Rademacher complexity term in (34) is being bounded.*

**Theorem 3.13.** *Under Assumption 3.11, for all $n \in \mathbb{Z}^+$ we have the Rademacher complexity bound*

$$\mathcal{R}_{\Psi,P,n} \leq 12n^{-1/2} E_P[L^2]^{1/2} \int_0^{D_\Theta} \sqrt{\log N(\epsilon, \Theta, d_\Theta)} d\epsilon. \tag{52}$$

*where $D_\Theta := \sup_{\theta_1, \theta_2 \in \Theta} d_\Theta(\theta_1, \theta_2)$ is the diameter of $\Theta$ and $N(\epsilon, \Theta, d_\Theta)$ denotes the covering number of $\Theta$ by $\epsilon$-balls in the metric $d_\Theta$.*

*In particular, if $\Theta$ is the unit ball in $\mathbb{R}^k$ under some norm then the entropy integral is finite and we have*

$$\mathcal{R}_{\Psi,P,n} \leq 48(k/n)^{1/2} E_P[L^2]^{1/2} = O((k/n)^{1/2}). \tag{53}$$

*Proof.* Using Dudley's entropy integral, see, e.g., Corollary 5.25 in van Handel (2016) or Theorem 5.22 in Wainwright (2019), we can bound the empirical Rademacher complexity at a sample $y \in \mathcal{Y}^n$ by

$$\widehat{\mathcal{R}}_{\Psi,n}(y) \leq 12n^{-1/2} \int_0^{D_n(y)} \sqrt{\log N(\epsilon, \Psi_n(y), \|\cdot\|_{L^2(n)})} d\epsilon, \tag{54}$$

where $N(\epsilon, \Psi_n(y), \|\cdot\|_{L^2(n)})$ is the $\epsilon$-covering number of $\Psi_n(y) := \{(\psi(y_1), ..., \psi(y_n)) : \psi \in \Psi\}$ under the norm $\|t\|_{L^2(n)} := \sqrt{\frac{1}{n}\sum_{i=1}^n t_i^2}$ and $D_n(y)$ is the diameter of $\Psi_n(y)$ under this norm.

The map $\Theta \to \Psi_n(y)$, $\theta \mapsto (\psi(\theta, y_1), ..., \psi(\theta, y_n))$ is onto and is $L_n(y) := \left(\frac{1}{n}\sum_{i=1}^n L(y_i)^2\right)^{1/2}$-Lipschitz with respect to $(d_\Theta, \|\cdot\|_{L^2(n)})$, as demonstrated by the calculation

$$\|(\psi(\theta_1, y_1), ..., \psi(\theta_1, y_n)) - (\psi(\theta_2, y_1), ..., \psi(\theta_2, y_n))\|_{L^2(n)}^2 = \frac{1}{n}\sum_{i=1}^n (\psi(\theta_1, y_i) - \psi(\theta_2, y_i))^2 \tag{55}$$

$$\leq \frac{1}{n}\sum_{i=1}^n L(y_i)^2 d_\Theta(\theta_1, \theta_2)^2.$$

These properties imply the following relation between covering numbers,

$$N(\epsilon, \Psi_n(y), \|\cdot\|_{L^2(n)}) \leq N(\epsilon/L_n(y), \Theta, d_\Theta), \tag{56}$$

as well as the diameter bounds $D_n(y) \leq L_n(y)D_\Theta$, where $D_\Theta := \sup_{\theta_1, \theta_2 \in \Theta} d_\Theta(\theta_1, \theta_2)$ is the diameter of $\Theta$. Combining these pieces and changing variables in the integral we arrive at

$$\widehat{\mathcal{R}}_{\Psi,n}(y) \leq 12n^{-1/2} \int_0^{L_n(y)D_\Theta} \sqrt{\log N(\epsilon/L_n(y), \Theta, d_\Theta)} d\epsilon \tag{57}$$

$$= 12n^{-1/2} L_n(y) \int_0^{D_\Theta} \sqrt{\log N(\epsilon, \Theta, d_\Theta)} d\epsilon.$$

Taking the expectation of both sides, we obtain at the Rademacher complexity bound

$$\mathcal{R}_{\Psi,P,n} \leq 12n^{-1/2} E_{P^n}[L_n] \int_0^{D_\Theta} \sqrt{\log N(\epsilon, \Theta, d_\Theta)} d\epsilon \,,$$

where

$$E_{P^n}[L_n] \leq \left( E_{P^n} \left[ \frac{1}{n} \sum_{i=1}^n L(y_i)^2 \right] \right)^{1/2} = E_P[L^2]^{1/2} \,. \tag{58}$$

If $\Theta$ is the unit ball in $\mathbb{R}^k$ with respect to the norm $\|\cdot\|_\Theta$ then $D_\Theta \leq 2$ and we have the covering number bound $N(\epsilon, \Theta, \|\cdot\|_\Theta) \leq (1 + 2/\epsilon)^k$, see, e.g., Example 5.8 in Wainwright (2019). Therefore

$$\int_0^{D_\Theta} \sqrt{\log N(\epsilon, \Theta, \|\cdot\|_\Theta)} d\epsilon \leq k^{1/2} \int_0^2 \sqrt{\log(1 + 2/\epsilon)} d\epsilon \leq \sqrt{2} k^{1/2} \int_0^2 \epsilon^{-1/2} d\epsilon = 4k^{1/2} \,. \tag{59}$$

Thus we arrive at (53). □

The Lipschitz property from Assumption 3.11 with $L(y) = a + b\|y\|$, $y \in \mathbb{R}^d$ holds for many neural network architectures, $\psi(\theta, y)$, parameterized by $\theta \in \Theta \subset \mathbb{R}^k$. In such cases, Theorem 3.13 implies $O(n^{-1/2})$ (resp. $O(m^{-1/2})$) Rademacher complexity bounds for GANs whenever the discriminator (resp. and generator) spaces are appropriate neural networks, assuming that $Q$ and $P_Z$ have finite second moments respectively. When combined with Theorem 3.7, this implies statistical consistency of the corresponding $(f, \Gamma)$-GANs. We emphasize that existence of the second moment is a much weaker assumption than the sub-exponential or sub-Gaussian requirements of previous approaches, and thus our Theorem 3.13 improves on the state of the art bounds even in the IPM-GAN case, i.e., Theorem 3.7 with $f^*(z) = z$. We also note that in cases where $P \sim Y + Z$ where $Y$ has compact support and $Z$ is a (possibly unbounded) perturbation with mean zero with $O(\delta)$ variance then Theorem 3.13 yields a Rademacher complexity bound that differs from the distribution-independent bound in the $Z = 0$ case by a $O((\delta k/n)^{1/2})$ term; thus, perturbing the data with a unbounded noise that has small variance results in a negligible difference in the statistical guarantees for all $n$.

## 4 Conclusion

We have derived statistical error bounds for $(f, \Gamma)$-GANs, a large class of GANs with nonlinear objective functionals. These GANs are based on the $(f, \Gamma)$-divergences, which generalize and interpolate between integral probability metrics (IPMs, e.g., 1-Wasserstein) and $f$-divergences (e.g., JS, KL, $\alpha$-divergences) and have been show to outperform both IPM and $f$-divergence-based methods in a number of applications. This paper extends earlier techniques for proving consistency of GANs with linear objective functionals (IPM-GANs) to the nonlinear objective setting. The key technical results are the tight perturbation bound in Lemma 2.7, uniform law of large numbers bounds for a class of nonlinear functionals in Lemmas 2.9 and 2.10, and the $(f, \Gamma)$-GAN error decompositions in Section 3.1. These results allow for the derivation of finite-sample concentration inequalities for $(f, \Gamma)$-GANs in Theorem 3.7. We also presented a new Rademacher complexity bound in Section 3.3 that implies the statistical consistency of $(f, \Gamma)$-GANs for distributions with unbounded support that have finite second moment. In particular, our results can be applied to certain heavy-tailed distributions, where the MGF does not exist, thus providing new insight even in the previously studied IPM-GAN case. Moreover, these novel Rademacher complexity bounds are of independent interest for analyzing other statistical learning tasks.

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

# A    Properties of $f$ and $f^*$

In this appendix we collect a number of important properties of the convex function $f$ and its Legendre transform $f^*$ that are needed in the study of $(f, \Gamma)$-GANs. For $a, b$ satisfying $-\infty \leq a < 1 < b \leq \infty$ we define $\mathcal{F}_1(a, b)$ to be the set of convex $f : (a, b) \to \mathbb{R}$ with $f(1) = 0$. For $f \in \mathcal{F}_1(a, b)$, standard convex functions theory, see, e.g., Rockafellar (1970) and Appendix A in Birrell et al. (2022b), implies that $f$ and its Legendre transform $f^* : \mathbb{R} \to (-\infty, \infty]$, $f^*(z) = \sup_{t \in (a,b)}\{zt - f(t)\}$, have the following properties:

1. $f^*(z) \geq z$ for all $z \in \mathbb{R}$ (this uses $f(1) = 0$).

2. If $a \geq 0$ then $f^*$ is non-decreasing.

3. $(f^*)^* = f$.

4. $f^*$ is convex and LSC.

5. $f^*$ is continuous on $\overline{\{f^* < \infty\}}$, where $\overline{A}$ denotes the closure of a set $A$.

6. The right derivative $(f^*)'_+$ exists and is finite on $\{f^* < \infty\}^o$, where $A^o$ denotes the interior of a set $A$. Similarly, $f'_+$ exists and is finite on $(a, b)$.

7. $(f^*)'_+$ is non-decreasing and absolutely continuous on compact intervals; the latter implies that $f^*$ satisfies the fundamental theorem of calculus, see, e.g., Theorem 3.35 and exercise 42 in Folland (2013).

Of particular relevance will be the value $z_0 := f'_+(1)$, which exists and is finite due to the assumption that $1 \in (a, b)$. The importance of $z_0$ to the $(f, \Gamma)$-divergences was observed in Birrell et al. (2022b), where the following properties were proven (see Lemma A.9):

**Lemma A.1.** *Define $z_0 := f'_+(1)$.*

1. $f^*(z_0) = z_0$

2. *If $f$ is strictly convex on a neighborhood of 1 and $z_0 \in \{f^* < \infty\}^o$ then $(f^*)'_+(z_0) = 1$.*

These properties will be key to our analysis and so we will generally assume that $f$ is strictly convex on a neighborhood of 1 and $z_0 \in \{f^* < \infty\}^o$.

# B  Properties of the $(f, \Gamma)$-Divergences

In this appendix we collect several important properties of the $(f, \Gamma)$-divergences. For $\mathcal{X}$ a measurable space we let $\mathcal{M}_b(\mathcal{X})$ denote the space of bounded measurable real-valued functions on $\mathcal{X}$ and $\mathcal{P}(\mathcal{X})$ be the set of probability measures on $\mathcal{X}$. The following results are taken from Theorem 2.8 of Birrell et al. (2022b).

**Theorem B.1.** *Let $f \in \mathcal{F}_1(a, b)$, $\Gamma \subset \mathcal{M}_b(\mathcal{X})$ be nonempty, and $Q, P \in \mathcal{P}(\mathcal{X})$.*

*1.*

$$D_f^\Gamma(Q\|P) \leq \inf_{\eta \in \mathcal{P}(\mathcal{X})} \{D_f(\eta\|P) + d_\Gamma(Q, \eta)\}. \tag{60}$$

*In particular, $D_f^\Gamma(Q\|P) \leq \min\{D_f(Q\|P), d_\Gamma(Q, P)\}$.*

*2. The map $(Q, P) \in \mathcal{P}(S) \times \mathcal{P}(S) \mapsto D_f^\Gamma(Q\|P)$ is convex.*

*3. If there exists $c_0 \in \Gamma \cap \mathbb{R}$ then $D_f^\Gamma(Q\|P) \geq 0$.*

*4. Suppose $f$ and $\Gamma$ satisfy the following:*

*(a) There exist a nonempty set $\Psi \subset \Gamma$ with the following properties:*
   *i. $\Psi$ is $\mathcal{P}(\mathcal{X})$-determining, i.e., for all $Q, P \in \mathcal{P}(\mathcal{X})$, $E_Q[\psi] = E_P[\psi]$ for all $\psi \in \Psi$ implies $Q = P$.*
   *ii. For all $\psi \in \Psi$ there exists $c_0 \in \mathbb{R}$, $\epsilon_0 > 0$ such that $c_0 + \epsilon\psi \in \Gamma$ for all $|\epsilon| < \epsilon_0$.*

*(b) $f$ is strictly convex on a neighborhood of $1$.*

*(c) $f^*$ is finite and $C^1$ on a neighborhood of $f'_+(1)$.*

*Then $D_f^\Gamma$ has the divergence property, i.e., $D_f^\Gamma(Q\|P) \geq 0$ with equality if and only if $Q = P$.*

**Remark B.2.** *Under stronger assumptions one can show that (60) is in fact an equality; see Theorem 2.15 in Birrell et al. (2022b).*

**Remark B.3.** *Assumptions 4(b) and 4(c) hold, for instance, if $f$ is strictly convex on $(a, b)$ and $f'_+(1) \in \{f^* < \infty\}^o$; see Theorem 26.3 in Rockafellar (1970).*

# C  Rademacher Complexity and Uniform Law of Large Numbers

In this appendix we recall the definition of Rademacher complexity and its use in proving uniform law of large numbers (ULLN) results. There are two definitions of Rademacher complexity in common use, those being with and without absolute value. In this work we use the version without absolute value as defined below.

**Definition C.1.** *Let $\Psi$ be a collection of functions on $\mathcal{Y}$ and $n \in \mathbb{Z}^+$. The empirical Rademacher complexity of $\Psi$ at $y \in \mathcal{Y}^n$ is defined by*

$$\widehat{\mathcal{R}}_{\Psi,n}(y) := E_\sigma \left[ \sup_{\psi \in \Psi} \left\{ \frac{1}{n} \sigma \cdot \psi_n(y) \right\} \right], \tag{61}$$

*where $\sigma_i$, $i = 1, ..., n$ are independent uniform random variables taking values in $\{-1, 1\}$, i.e., Rademacher random variables, and $\psi_n(y) := (\psi(y_1), ..., \psi(y_n))$. Given a probability distribution $P$ on $\mathcal{Y}$, the Rademacher complexity of $\Psi$ relative to $P$ is defined by*

$$\mathcal{R}_{\Psi,P,n} := E_{P^n} \left[ \widehat{\mathcal{R}}_{\Psi,n} \right] \tag{62}$$

*where $P^n$ is the $n$-fold product of $P$, i.e., the $y_i$'s become i.i.d. samples from $P$.*

**Remark C.2.** *Note that in (62) we are implicitly assuming that the empirical Rademacher complexity is measurable. When we use (62) in the main text, this will be guaranteed by other assumptions.*

The definition (61) is particularly convenient for our purposes due to the following version of Talagrand's lemma; see Lemma 5.7 in Mohri et al. (2018).

**Lemma C.3.** *Let $\alpha, \beta \in \mathbb{R}$ and $\Psi$ be a collection of real-valued functions on $\mathcal{Y}$ such that $\alpha \leq \psi \leq \beta$ for all $\psi \in \Psi$. If $\phi : [\alpha, \beta] \to \mathbb{R}$ is $L$-Lipschitz then*

$$\widehat{\mathcal{R}}_{\phi \circ \Psi, n}(y) \leq L \widehat{\mathcal{R}}_{\Psi, n}(y) \tag{63}$$

*for all $n \in \mathbb{Z}^+$, $y \in \mathcal{Y}^n$, where $\phi \circ \Psi := \{\phi \circ \psi : \psi \in \Psi\}$.*

Bounds on the Rademacher complexity can be used to prove ULLN bounds; see, e.g., Theorem 3.3, Eq. (3.8) - (3.13) in Mohri et al. (2018) (this reference assumes $[0, 1]$-valued functions but the result can be freely shifted and scaled to apply to a set of uniformly bounded functions):

**Theorem C.4** (ULLN). *Let $P$ be a probability measure on $\mathcal{Y}$ and $\Psi$ a countable family of real-valued measurable functions on $\mathcal{Y}$ that are uniformly bounded. If $Y_i$, $i = 1, ..., n$ are i.i.d. $\mathcal{Y}$-valued random variables that are distributed as $P$ then for $n \in \mathbb{Z}^+$ we have*

$$\mathbb{E}\left[ \sup_{\psi \in \Psi} \left\{ \pm \left( \frac{1}{n} \sum_{i=1}^n \psi(Y_i) - E_P[\psi] \right) \right\} \right] \leq 2 \mathcal{R}_{\Psi, P, n}. \tag{64}$$

**Remark C.5.** *We have stated the ULLN in terms of a countable collection of functions to avoid measurability issues, but under appropriate assumptions one can apply it to an uncountable $\Psi$, e.g., if there exists a countable $\Psi_0 \subset \Psi$ such that for every $\psi \in \Psi$ there exists a sequence $\psi_j \in \Psi_0$ with $\psi_j \to \psi$ pointwise.*

# D  Proofs of Important Lemmas

In this appendix we present the proofs of several important lemmas from the main text. We begin by deriving the properties of $\Lambda_f^P$ that were stated in Section 2.

*Proof of Lemma 2.1.*      1. $f^*$ is non-decreasing, therefore

$$\inf_{\nu \in \mathbb{R}} \{\nu + f^*(\alpha - \nu)\} \leq \inf_{\nu \in \mathbb{R}} \{\nu + E_P[f^*(h - \nu)]\} \leq \inf_{\nu \in \mathbb{R}} \{\nu + f^*(\beta - \nu)\} \tag{65}$$

for all $\nu$. For any $c \in \mathbb{R}$ we have

$$\inf_{\nu \in \mathbb{R}} \{\nu + f^*(c - \nu)\} = \inf_z \{c - z + f^*(c - (c - z))\} \tag{66}$$
$$= c - \sup_{z \in \mathbb{R}} \{z - f^*(z)\} = c - (f^*)^*(1) = c - f(1) = c.$$

Applying this to $c = \alpha$ and $c = \beta$ we see that $\alpha \leq \Lambda_f^P[h] \leq \beta$ as claimed.

2. For the following it will be useful to recall that $f^*(z_0) = z_0$ and $(f^*)'_+(z_0) = 1$; see Lemma A.1. Suppose $\nu > \beta - z_0$ and first consider the case where $f^*(h - \nu) < \infty$. We have $h - \nu < h - (\beta - z_0) \leq z_0$, therefore $h - \nu + 1/n, h - (\beta - z_0) \in \{f^* < \infty\}^o$ for $n$ sufficiently large. $f^*$ is absolutely continuous on compact intervals, therefore the fundamental theorem of calculus holds, see, e.g., Theorem 3.35 and exercise 42 in Folland (2013), and hence we have

$$f^*(h - (\beta - z_0)) = f^*(h - \nu + 1/n) + \int_{h - \nu + 1/n}^{h - (\beta - z_0)} (f^*)'_+(z) dz. \tag{67}$$

For $z \leq h - (\beta - z_0)$ we have $z \leq z_0$, hence $(f^*)'_+(z) \leq (f^*)'_+(z_0) = 1$. Therefore

$$f^*(h - (\beta - z_0)) \leq f^*(h - \nu + 1/n) + h - (\beta - z_0) - (h - \nu + 1/n) \tag{68}$$
$$= f^*(h - \nu + 1/n) - (\beta - z_0) + \nu - 1/n.$$

Using the continuity of $f^*$ on $\{f^* < \infty\}$ we can take $n \to \infty$ to get

$$\nu + f^*(h - \nu) \geq (\beta - z_0) + f^*(h - (\beta - z_0)). \tag{69}$$

This was proven under the assumption that $f^*(h - \nu) < \infty$, but it also trivially holds when this is infinite. Taking the expectation of both sides we therefore find

$$\nu + E_P[f^*(h - \nu)] \geq (\beta - z_0) + E_P[f^*(h - (\beta - z_0))] \geq \inf_{\nu \in [\alpha - z_0, \beta - z_0]} \{\nu + E_P[f^*(h - \nu)]\} \tag{70}$$

for all $\nu > \beta - z_0$.

Now suppose $\nu < \alpha - z_0$ and first consider the case where $f^*(h - \nu) < \infty$. We have $h - \nu > h - (\alpha - z_0) \geq z_0$. Therefore $h - \nu - 1/n, h - (\alpha - z_0) \in \{f^* < \infty\}^o$ for $n$ sufficiently large and

$$f^*(h - \nu - 1/n) = f^*(h - (\alpha - z_0)) + \int_{h - (\alpha - z_0)}^{h - \nu - 1/n} (f^*)'_+(z) dz. \tag{71}$$

For $z \geq h - (\alpha - z_0)$ we have $z \geq z_0$ and so $(f^*)'_+(z) \geq 1$. Hence

$$\begin{aligned} f^*(h - \nu - 1/n) \geq& f^*(h - (\alpha - z_0)) + h - \nu - 1/n - (h - (\alpha - z_0)) \\ =& f^*(h - (\alpha - z_0)) - \nu - 1/n + (\alpha - z_0). \end{aligned} \tag{72}$$

Taking $n \to \infty$ gives

$$\nu + f^*(h - \nu) \geq (\alpha - z_0) + f^*(h - (\alpha - z_0)). \tag{73}$$

This was proven under the assumption that $f^*(h - \nu) < \infty$, but it also trivially holds when this is infinite. Taking the expectation of both sides we therefore find

$$\nu + E_P[f^*(h - \nu)] \geq (\alpha - z_0) + E_P[f^*(h - (\alpha - z_0))] \geq \inf_{\nu \in [\alpha - z_0, \beta - z_0]} \{\nu + E_P[f^*(h - \nu)]\} \tag{74}$$

for all $\nu < \alpha - z_0$. Therefore we conclude that

$$\Lambda_f^P[h] = \inf_{\nu \in [\alpha - z_0, \beta - z_0]} \{\nu + E_P[f^*(h - \nu)]\} \tag{75}$$

as claimed.

$\square$

*Proof of Lemma 2.3.* $f^*$ is non-decreasing, hence $z_0 + \beta - \alpha \in \{f^* < \infty\}^o$ implies $z_0 \in \{f^* < \infty\}^o$. Therefore we can use Lemma 2.1 to write

$$\Lambda_f^P[h] = \inf_{\nu \in [\alpha - z_0, \beta - z_0]} \{\nu + E_P[f^*(h - \nu)]\}. \tag{76}$$

The fact that $f^*$ is non-decreasing also implies that $h - z_0 \leq f^*(h - \nu) \leq f^*(\beta - \alpha + z_0) < \infty$ for all $\nu \in [\alpha - z_0, \beta - z_0]$ and hence $E_P[f^*(h - \nu)]$ is finite. The same holds for $\tilde{h}$, therefore

$$|\Lambda_f^P[\tilde{h}] - \Lambda_f^P[h]| \leq \sup_{\nu \in [\alpha - z_0, \beta - z_0]} E_P[|f^*(\tilde{h} - \nu) - f^*(h - \nu)|]. \tag{77}$$

Here we made use of the simple bound

$$\pm(\inf_{i \in I} d_i - \inf_{i \in I} c_i) \leq \sup_{i \in I} \{\pm(d_i - c_i)\} \tag{78}$$

whenever $c_i, d_i \in \mathbb{R}$ for all $i \in I$ and $\inf_i c_i, \inf_i d_i$ are finite.

$f^*$ is absolutely continuous on the interval between $h - \nu$ and $\tilde{h} - \nu$, therefore we can use the fundamental theorem of calculus to write

$$f^*(\tilde{h} - \nu) - f^*(h - \nu) = \int_{h-\nu}^{\tilde{h}-\nu} (f^*)'_+(z)dz \tag{79}$$

and so

$$|f^*(\tilde{h} - \nu) - f^*(h - \nu)| \leq |\tilde{h} - h| \sup_{z \in [z_0 - (\beta - \alpha), z_0 + \beta - \alpha]} |(f^*)'_+(z)|. \tag{80}$$

The fact that $f^*$ is non-decreasing implies $(f^*)'_+ \geq 0$ on $\{f^* < \infty\}^o$. Combined with the fact that $(f^*)'_+$ is non-decreasing, we can therefore conclude

$$|f^*(\tilde{h} - \nu) - f^*(h - \nu)| \leq |\tilde{h} - h|(f^*)'_+(z_0 + \beta - \alpha). \tag{81}$$

Taking the expectation of both sides and combining the result with (77) we obtain the claimed Lipschitz bound

$$|\Lambda_f^P[\tilde{h}] - \Lambda_f^P[h]| \leq (f^*)'_+(z_0 + \beta - \alpha)\|\tilde{h} - h\|_{L^1(P)}. \tag{82}$$

$\square$

*Proof of Lemma 2.7.* First let $x_i = \alpha$ for all $i$, $\tilde{x}_i = \alpha$ for $i \neq j$ and $\tilde{x}_j = \beta$. We have $(x, \tilde{x}) \in E$ and therefore

$$\sup_{(x,\tilde{x}) \in E} \{\Lambda_f(\tilde{x}) - \Lambda_f(x)\} \tag{83}$$

$$\geq \inf_{\nu \in [\alpha - z_0, \beta - z_0]} \left\{\nu + \frac{1}{n}\sum_{i=1}^n f^*(\tilde{x}_i - \nu)\right\} - \inf_{\nu \in \mathbb{R}} \left\{\nu + \frac{1}{n}\sum_{i=1}^n f^*(x_i - \nu)\right\}$$

$$= \inf_{\nu \in [\alpha - z_0, \beta - z_0]} \left\{\nu - \alpha + \frac{n-1}{n}f^*(\alpha - \nu) + \frac{1}{n}f^*(\beta - \nu)\right\} + \sup_{\nu \in \mathbb{R}}\{\alpha - \nu - f^*(\alpha - \nu)\}$$

$$= \inf_{z \in [z_0 - (\beta - \alpha), z_0]} \left\{-z + \frac{n-1}{n}f^*(z) + \frac{1}{n}f^*(\beta - \alpha + z)\right\} + (f^*)^*(1)$$

$$= \inf_{z \in [z_0 - (\beta - \alpha), z_0]} \left\{-z + \frac{n-1}{n}f^*(z) + \frac{1}{n}f^*(\beta - \alpha + z)\right\}.$$

To prove the reverse inequality, let $(x, \tilde{x}) \in E$ be arbitrary and compute

$$\Lambda_f(\tilde{x}) - \Lambda_f(x) \tag{84}$$

$$= \inf_{\tilde{\nu} \in [\alpha - z_0, \beta - z_0]} \left\{\tilde{\nu} + \frac{1}{n}\sum_{i=1}^n f^*(\tilde{x}_i - \tilde{\nu})\right\} - \inf_{\nu \in [\alpha - z_0, \beta - z_0]} \left\{\nu + \frac{1}{n}\sum_{i=1}^n f^*(x_i - \nu)\right\}$$

$$= \sup_{\nu \in [\alpha - z_0, \beta - z_0]} \inf_{\tilde{\nu} \in [\alpha - z_0, \beta - z_0]} \left\{\tilde{\nu} - \nu + \frac{1}{n}\sum_{i=1, i \neq j}^n f^*(x_i - \tilde{\nu}) - \frac{1}{n}\sum_{i=1, i \neq j}^n f^*(x_i - \nu)\right.$$

$$\left. + \frac{1}{n}f^*(\tilde{x}_j - \tilde{\nu}) - \frac{1}{n}f^*(x_j - \nu)\right\}$$

$$\leq \sup_{\nu \in [\alpha - z_0, \beta - z_0]} \inf_{\tilde{\nu} \in [\alpha - z_0, \beta - z_0]} \left\{\tilde{\nu} - \nu + \frac{1}{n}\sum_{i=1, i \neq j}^n f^*(x_i - \tilde{\nu}) - \frac{1}{n}\sum_{i=1, i \neq j}^n f^*(x_i - \nu)\right.$$

$$\left. + \frac{1}{n}f^*(\beta - \tilde{\nu}) - \frac{1}{n}f^*(\alpha - \nu)\right\},$$

where in the last line we used the fact that $f^*$ is non-decreasing. We have $\tilde{x}_i - \tilde{\nu}, x_i - \nu \in \{f^* < \infty\}^o$ and for each $i \neq j$ the terms involving $x_i$ are absolutely continuous on compact intervals with right derivative

$$\frac{d}{dx_i^+} \left(\frac{1}{n}f^*(x_i - \tilde{\nu}) - \frac{1}{n}f^*(x_i - \nu)\right) = \frac{1}{n}\left((f^*)'_+(x_i - \tilde{\nu}) - (f^*)'_+(x_i - \nu)\right), \tag{85}$$

which is non-positive when $\tilde{\nu} \geq \nu$ and non-negative when $\tilde{\nu} \leq \nu$ (due to $(f^*)'_+$ being non-decreasing). The fundamental theorem of calculus then implies that $\frac{1}{n}f^*(x_i - \tilde{\nu}) - \frac{1}{n}f^*(x_i - \nu)$ is non-increasing in $x_i$ when $\tilde{\nu} \geq \nu$ and non-decreasing in $x_i$ when $\tilde{\nu} \leq \nu$. Therefore

$$\Lambda_f(\tilde{x}) - \Lambda_f(x) \tag{86}$$

$$\leq \sup_{\nu \in [\alpha - z_0, \beta - z_0]} \inf_{\tilde{\nu} \in [\alpha - z_0, \beta - z_0]} \begin{cases} \tilde{\nu} - \nu + f^*(\beta - \tilde{\nu}) - \frac{n-1}{n}f^*(\beta - \nu) - \frac{1}{n}f^*(\alpha - \nu) & \text{if } \tilde{\nu} < \nu \\ \tilde{\nu} - \nu + \frac{n-1}{n}f^*(\alpha - \tilde{\nu}) - f^*(\alpha - \nu) + \frac{1}{n}f^*(\beta - \tilde{\nu}) & \text{if } \tilde{\nu} \geq \nu \end{cases}$$

$$= \sup_{\nu \in [\alpha - z_0, \beta - z_0]} \inf_{z \in [z_0 - (\beta - \alpha), z_0]} \begin{cases} \alpha - z - \nu + f^*(\beta - \alpha + z) - \frac{n-1}{n}f^*(\beta - \nu) - \frac{1}{n}f^*(\alpha - \nu) & \text{if } \alpha - \nu < z \\ \alpha - z - \nu + \frac{n-1}{n}f^*(z) - f^*(\alpha - \nu) + \frac{1}{n}f^*(\beta - \alpha + z) & \text{if } \alpha - \nu \geq z \end{cases},$$

where we changed variables to $z = \alpha - \tilde{\nu}$ in the last line. For $z \in (\alpha - \nu, z_0]$, the right derivative of the objective with respect to $z$ is given by

$$\frac{d}{dz^+}\left( \alpha - z - \nu + f^*(\beta - \alpha + z) - \frac{n-1}{n}f^*(\beta - \nu) - \frac{1}{n}f^*(\alpha - \nu) \right) = -1 + (f^*)'_+(\beta - \alpha + z). \tag{87}$$

$(f^*)'_+$ is non-decreasing and $\beta - \alpha + z \geq z_0$, therefore $-1 + (f^*)'_+(\beta - \alpha + z) \geq -1 + (f^*)'_+(z_0) = 0$. This implies that the objective is non-decreasing in $z \in [\alpha - \nu, z_0]$ (the extension to the endpoint $\alpha - \nu$ follows from continuity of the objective). Therefore

$$\Lambda_f(\tilde{x}) - \Lambda_f(x) \tag{88}$$

$$\leq \sup_{\nu \in [\alpha - z_0, \beta - z_0]} \inf_{z \in [z_0 - (\beta - \alpha), \alpha - \nu]} \left\{ \alpha - z - \nu + \frac{n-1}{n}f^*(z) - f^*(\alpha - \nu) + \frac{1}{n}f^*(\beta - \alpha + z) \right\}$$

$$= \sup_{\nu \in [\alpha - z_0, \beta - z_0]} \left\{ -\nu - f^*(\alpha - \nu) + \inf_{z \in [z_0 - (\beta - \alpha), \alpha - \nu]} \left\{ \alpha - z + \frac{n-1}{n}f^*(z) + \frac{1}{n}f^*(\beta - \alpha + z) \right\} \right\}.$$

Let $z_*$ be a minimizer of

$$\inf_{z \in [z_0 - (\beta - \alpha), z_0]} \left\{ \alpha - z + \frac{n-1}{n}f^*(z) + \frac{1}{n}f^*(\beta - \alpha + z) \right\}, \tag{89}$$

which exists due to compactness of the domain and continuity of the objective. The objective is convex in $z$, therefore it is non-increasing on $(-\infty, z_*]$ and so

$$\Lambda_f(\tilde{x}) - \Lambda_f(x) \tag{90}$$

$$\leq \sup_{\nu \in [\alpha - z_0, \beta - z_0]} \left\{ -\nu - f^*(\alpha - \nu) + \begin{cases} \alpha - z_* + \frac{n-1}{n}f^*(z_*) + \frac{1}{n}f^*(\beta - \alpha + z_*) & \text{if } \alpha - \nu \geq z_* \\ \nu + \frac{n-1}{n}f^*(\alpha - \nu) + \frac{1}{n}f^*(\beta - \nu) & \text{if } \alpha - \nu < z_* \end{cases} \right\}$$

$$= \sup_{w \in [z_0 - (\beta - \alpha), z_0]} \begin{cases} w - f^*(w) - z_* + \frac{n-1}{n}f^*(z_*) + \frac{1}{n}f^*(\beta - \alpha + z_*) & \text{if } w \geq z_* \\ \frac{1}{n}(f^*(\beta - \alpha + w) - f^*(w)) & \text{if } w < z_* \end{cases},$$

where we changed variables to $w = \alpha - \nu$. We have $\sup_{w \in \mathbb{R}}\{w - f^*(w)\} = (f^*)^*(1) = 0$ and $w \mapsto f^*(\beta - \alpha + w) - f^*(w)$ is non-decreasing on $w \leq z_0$ (this follows from $(f^*)'_+(\beta - \alpha + w) - (f^*)'_+(w) \geq 0$), therefore

$$\Lambda_f(\tilde{x}) - \Lambda_f(x) \leq \max\left\{ -z_* + \frac{n-1}{n}f^*(z_*) + \frac{1}{n}f^*(\beta - \alpha + z_*), \frac{1}{n}(f^*(\beta - \alpha + z_*) - f^*(z_*)) \right\}. \tag{91}$$

Using the bound $f^*(z) \geq z$ we can compute

$$-z_* + \frac{n-1}{n}f^*(z_*) + \frac{1}{n}f^*(\beta - \alpha + z_*) - \frac{1}{n}(f^*(\beta - \alpha + z_*) - f^*(z_*))$$
$$= -z_* + f^*(z_*) \geq 0.$$

Hence

$$\Lambda_f(\tilde{x}) - \Lambda_f(x) \leq -z_* + \frac{n-1}{n} f^*(z_*) + \frac{1}{n} f^*(\beta - \alpha + z_*) \tag{92}$$

$$= \inf_{z \in [z_0 - (\beta - \alpha), z_0]} \left\{ -z + \frac{n-1}{n} f^*(z) + \frac{1}{n} f^*(\beta - \alpha + z) \right\}.$$

This holds for all $(x, \tilde{x}) \in E$ and so when combined with (83) we obtain the claimed equality.

To derive the looser bounds, first use $f^*(z) \geq z$ to compute the lower bound

$$\inf_{z \in [z_0 - (\beta - \alpha), z_0]} \left\{ -z + \frac{n-1}{n} f^*(z) + \frac{1}{n} f^*(\beta - \alpha + z) \right\} \geq \frac{\beta - \alpha}{n}. \tag{93}$$

For the upper bound we compute

$$\inf_{z \in [z_0 - (\beta - \alpha), z_0]} \left\{ -z + \frac{n-1}{n} f^*(z) + \frac{1}{n} f^*(\beta - \alpha + z) \right\} \tag{94}$$

$$\leq -z_0 + \frac{n-1}{n} f^*(z_0) + \frac{1}{n} f^*(\beta - \alpha + z_0) = \frac{1}{n}(f^*(\beta - \alpha + z_0) - z_0)$$

$$\leq (f^*)'_+(\beta - \alpha + z_0) \frac{\beta - \alpha}{n},$$

where to obtain the last line we used the fundamental theorem of calculus together with the facts that $f^*(z_0) = z_0$ and $(f^*)'_+$ is non-decreasing. $\qquad\square$

*Proof of Lemma 2.9.* The result is trivial if $\alpha = \beta$ so suppose $\alpha < \beta$. Note that $f^*$ is non-decreasing, and hence $z_0 + \beta - \alpha \in \{f^* < \infty\}^o$ implies $z_0 \in \{f^* < \infty\}^o$. Using Lemma 2.1 along with (78) we obtain

$$\pm \left( \Lambda_f^P[\psi] - \Lambda_f^{P_n}[\psi] \right) \leq \sup_{\nu \in [\alpha - z_0, \beta - z_0]} \{ \pm(E_P[f^*(\psi - \nu)] - E_{P_n}[f^*(\psi - \nu)]) \}. \tag{95}$$

Note that $z_0 - (\beta - \alpha) \leq f^*(\psi - \nu) \leq f^*(\beta - \alpha + z_0) < \infty$ and so the expectations are finite. Also, the supremum over $\nu$ can be restricted to rational $\nu$ due to continuity. Maximizing over $\psi$ and taking expectations gives

$$\mathbb{E}\left[ \sup_{\psi \in \Psi} \left\{ \pm \left( \Lambda_f^P[\psi] - \Lambda_f^{P_n}[\psi] \right) \right\} \right] \leq \mathbb{E}\left[ \sup_{\psi \in \Psi, \nu \in [\alpha - z_0, \beta - z_0] \cap \mathbb{Q}} \{ \pm(E_P[f^*(\psi - \nu)] - E_{P_n}[f^*(\psi - \nu)]) \} \right]. \tag{96}$$

The ULLN (see Theorem C.4) implies

$$\mathbb{E}\left[ \sup_{\psi \in \Psi, \nu \in [\alpha - z_0, \beta - z_0] \cap \mathbb{Q}} \{ \pm(E_P[f^*(\psi - \nu)] - E_{P_n}[f^*(\psi - \nu)]) \} \right] \tag{97}$$

$$= \mathbb{E}\left[ \sup_{h \in \mathcal{H}} \left\{ \pm \left( E_P[h] - \frac{1}{n} \sum_{i=1}^n h(Y_i) \right) \right\} \right] \leq 2\mathcal{R}_{\mathcal{H}, P, n},$$

where $\mathcal{H} := \{f^*(\psi - \nu) : \psi \in \Psi, \nu \in [\alpha - z_0, \beta - z_0] \cap \mathbb{Q}\}$. $f^*$ and $(f^*)'$ are decreasing, therefore $f^*$ is $(f^*)'_+(\beta - \alpha + z_0)$-Lipschitz on $(-\infty, \beta - \alpha + z_0]$ and so Talagrand's lemma (see Lemma C.3) implies

$$\mathcal{R}_{\mathcal{H}, P, n} \leq (f^*)'_+(\beta - \alpha + z_0) \mathcal{R}_{\Psi - [\alpha - z_0, \beta - z_0] \cap \mathbb{Q}, P, n}. \tag{98}$$

We can compute

$$\mathcal{R}_{\Psi - [\alpha - z_0, \beta - z_0] \cap \mathbb{Q}, P, n} = \mathcal{R}_{\Psi, P, n} + \mathcal{R}_{[\alpha - z_0, \beta - z_0], P, n} \tag{99}$$

and

$$\mathcal{R}_{[\alpha-z_0,\beta-z_0],P,n} = \frac{1}{n} E_\sigma \left[ \sup_{\nu\in[\alpha-z_0,\beta-z_0]} \nu \sum_{i=1}^n \sigma_i \right] = \frac{1}{n} E_\sigma \left[ \sup_{z\in[-(\beta-\alpha)/2,(\beta-\alpha)/2]} (z + (\beta+\alpha)/2 - z_0) \sum_{i=1}^n \sigma_i \right]$$
(100)

$$= \frac{1}{n} E_\sigma \left[ \sup_{z\in[-(\beta-\alpha)/2,(\beta-\alpha)/2]} z \sum_{i=1}^n \sigma_i \right] = \frac{\beta-\alpha}{2n} E_\sigma \left[ \left| \sum_{i=1}^n \sigma_i \right| \right] \leq \frac{\beta-\alpha}{2n} E_\sigma \left[ \left| \sum_{i=1}^n \sigma_i \right|^2 \right]^{1/2}$$

$$= \frac{\beta-\alpha}{2n^{1/2}} .$$

Combining these completes the proof. $\qquad\square$

*Proof of Lemma 2.10.* As in the proof of Lemma 2.9, the result is trivial when $\alpha = \beta$. For $\alpha < \beta$ we have

$$\mathbb{E}\left[ \sup_{\psi\in\Psi} \left\{ \pm\left(\Lambda_f^P[\psi] - \Lambda_f^{P_n}[\psi]\right) \right\} \right] \leq \mathbb{E}\left[ \sup_{\psi\in\Psi,\nu\in[\alpha-z_0,\beta-z_0]\cap\mathbb{Q}} \left\{ \pm(E_P[f^*(\psi-\nu)] - E_{P_n}[f^*(\psi-\nu)]) \right\} \right] . \quad (101)$$

Now use the fundamental theorem of calculus to compute

$$f^*(\psi-\nu) = f^*(\psi-(\beta-z_0)) + (\beta-z_0-\nu)\int_0^1 (f^*)'_+(\psi-(1-t)(\beta-z_0)-t\nu)dt \quad (102)$$

for $\psi\in\Psi, \nu\in[\alpha-z_0,\beta-z_0]\cap\mathbb{Q}$. Therefore

$$\sup_{\psi\in\Psi,\nu\in[\alpha-z_0,\beta-z_0]\cap\mathbb{Q}} \left\{ \pm(E_P[f^*(\psi-\nu)] - E_{P_n}[f^*(\psi-\nu)]) \right\} \quad (103)$$

$$= \sup_{\psi\in\Psi,\nu\in[\alpha-z_0,\beta-z_0]\cap\mathbb{Q}} \left\{ \pm\left(E_P[f^*(\psi-(\beta-z_0))] - E_{P_n}[f^*(\psi-(\beta-z_0))] \right.\right.$$
$$\left.\left. + (\beta-z_0-\nu)\int_0^1 E_P[(f^*)'_+(\psi-(1-t)(\beta-z_0)-t\nu)] - E_{P_n}[(f^*)'_+(\psi-(1-t)(\beta-z_0-t\nu)]dt \right)\right\}$$

$$\leq \sup_{\psi\in\Psi}\left\{\pm(E_P[f^*(\psi-(\beta-z_0))] - E_{P_n}[f^*(\psi-(\beta-z_0))])\right\}$$

$$+ \sup_{\psi\in\Psi,\nu\in[\alpha-z_0,\beta-z_0]\cap\mathbb{Q}} \left\{ (\beta-z_0-\nu)\int_0^1 \left( E_P[(f^*)'_+(\psi-(1-t)(\beta-z_0)-t\nu)] \right.\right.$$
$$\left.\left. - E_{P_n}[(f^*)'_+(\psi-(1-t)(\beta-z_0-t\nu)] \right)dt \right\}$$

$$\leq \sup_{h\in f^*\circ(\Psi-(\beta-z_0))}\left\{\pm(E_P[h] - E_{P_n}[h]\right\} \quad (104)$$

$$+ \sup_{\psi\in\Psi,\nu\in[\alpha-z_0,\beta-z_0]\cap\mathbb{Q}} \left\{ (\beta-z_0-\nu)\int_0^1 \left( E_P[(f^*)'_+(\psi-(1-t)(\beta-z_0)-t\nu)] \right.\right.$$
$$\left.\left. - E_{P_n}[(f^*)'_+(\psi-(1-t)(\beta-z_0-t\nu)] \right)dt \right\} .$$

The ULLN (Theorem C.4) implies

$$\mathbb{E}\left[ \sup_{h\in f^*\circ(\Psi-(\beta-z_0))} \left\{\pm(E_P[h] - E_{P_n}[h])\right\} \right] \leq 2\mathcal{R}_{f^*\circ(\Psi-(\beta-z_0)),P,n} . \quad (105)$$

We have $\psi-(\beta-z_0) \leq z_0$ and $f^*$ is $(f^*)'_+(z_0)$-Lipschitz on $(-\infty,z_0]$ where $(f^*)'_+(z_0) = 1$, hence Talagrand's lemma (Lemma C.3) implies

$$\mathcal{R}_{f^*\circ(\Psi-(\beta-z_0)),P,n} \leq \mathcal{R}_{\Psi-(\beta-z_0),P,n} = \mathcal{R}_{\Psi,P,n} . \quad (106)$$

We note that the shift invariance of the Rademacher complexity follows from the definition, as recalled in Eq. (61) of Appendix C.

As for the second term in (104), we have

$$
\mathbb{E}\left[\sup_{\psi\in\Psi,\nu\in[\alpha-z_0,\beta-z_0]\cap\mathbb{Q}}\left\{(\beta-z_0-\nu)\int_0^1\left(E_P[(f^*)'_+(\psi-(1-t)(\beta-z_0)-t\nu)]\right.\right.\right.
$$
$$
\left.\left.\left.-E_{P_n}[(f^*)'_+(\psi-(1-t)(\beta-z_0-t\nu)]\right)dt\right\}\right]
$$
(107)

$$
\leq(\beta-\alpha)\int_0^1\mathbb{E}\left[\sup_{\psi\in\Psi,\nu\in[\alpha-z_0,\beta-z_0]\cap\mathbb{Q}}\left\{\left|E_P[(f^*)'_+(\psi-(1-t)(\beta-z_0)-t\nu)]\right.\right.\right.
$$
$$
\left.\left.\left.-E_{P_n}[(f^*)'_+(\psi-(1-t)(\beta-z_0-t\nu)]\right|\right\}\right]dt\,.
$$

Using the fact that $\Psi$ contains a constant (which implies that the terms in the following bound are non-negative) and then employing the ULLN (Theorem C.4) we have

$$
\mathbb{E}\left[\sup_{\psi\in\Psi,\nu\in[\alpha-z_0,\beta-z_0]\cap\mathbb{Q}}|E_P[(f^*)'_+(\psi-(1-t)(\beta-z_0)-t\nu)]-E_{P_n}[(f^*)'_+(\psi-(1-t)(\beta-z_0)-t\nu)]|\right]
$$
(108)

$$
\leq\mathbb{E}\left[\sup_{\psi\in\Psi,\nu\in[\alpha-z_0,\beta-z_0]\cap\mathbb{Q}}\{E_P[(f^*)'_+(\psi-(1-t)(\beta-z_0)-t\nu)]-E_{P_n}[(f^*)'_+(\psi-(1-t)(\beta-z_0)-t\nu)]\}\right]
$$

$$
+\mathbb{E}\left[\sup_{\psi\in\Psi,\nu\in[\alpha-z_0,\beta-z_0]\cap\mathbb{Q}}\{-(E_P[(f^*)'_+(\psi-(1-t)(\beta-z_0)-t\nu)]-E_{P_n}[(f^*)'_+(\psi-(1-t)(\beta-z_0)-t\nu)])\}\right]
$$

$$
\leq4\mathcal{R}_{\mathcal{H}_t,P,n}\,,
$$

where $\mathcal{H}_t:=\{(f^*)'_+(\psi-(1-t)(\beta-z_0)-t\nu):\psi\in\Psi,\nu\in[\alpha-z_0,\beta-z_0]\cap\mathbb{Q}\}$. Using the Lipschitz assumption on $(f^*)'_+$ together with Talagrand's lemma (Lemma C.3) and the result of the calculation (100) we obtain

$$
\mathcal{R}_{\mathcal{H}_t,P,n}\leq L_{\alpha,\beta}\mathcal{R}_{\Psi-(1-t)(\beta-z_0)-t[\alpha-z_0,\beta-z_0]\cap\mathbb{Q}),P,n}
$$
(109)
$$
=L_{\alpha,\beta}(\mathcal{R}_{\Psi,P,n}+tR_{[\alpha-z_0,\beta-z_0]\cap\mathbb{Q}),P,n})\leq L_{\alpha,\beta}\left(\mathcal{R}_{\Psi,P,n}+t\frac{\beta-\alpha}{2n^{1/2}}\right)\,.
$$

Putting these bounds together we find

$$
\mathbb{E}\left[\sup_{\psi\in\Psi}\left\{\pm\left(\Lambda_f^P[\psi]-\Lambda_f^{P_n}[\psi]\right)\right\}\right]\leq2\mathcal{R}_{\Psi,P,n}+(\beta-\alpha)\int_0^14L_{\alpha,\beta}\left(\mathcal{R}_{\Psi,P,n}+t\frac{\beta-\alpha}{2n^{1/2}}\right)dt
$$
(110)
$$
=2(1+2(\beta-\alpha)L_{\alpha,\beta})\mathcal{R}_{\Psi,P,n}+\frac{(\beta-\alpha)^2L_{\alpha,\beta}}{n^{1/2}}\,.
$$

Combining this with the bound from Lemma 2.9 gives the claimed result. $\qquad\square$

Finally, we derive the alternative error decomposition from Lemma 3.5.

*Proof of Lemma 3.5.* Using the assumption (23) along with Lemma 3.1 we can compute the $\mathbb{P}$-a.s. bound

$$
D_f^\Gamma(Q\|P_{\theta^*_{n,m}})\leq D_f^{\widetilde{\Gamma}}(Q\|P_{\theta^*_{n,m}})+\inf_{\theta\in\Theta}D_f^{\widetilde{\Gamma}}(Q_n\|P_{\theta,m})-D_f^{\widetilde{\Gamma}}(Q_n\|P_{\theta^*_{n,m},m})
$$
(111)
$$
+\left(1+(f^*)'_+(z_0+\beta-\alpha)\right)\sup_{h\in\Gamma}\inf_{\tilde{h}\in\widetilde{\Gamma}}\|h-\tilde{h}\|_\infty+\epsilon_{n,m}^{opt}\,.
$$

As a simple consequence of the definition (3), for all $\theta \in \Theta$ we have

$$D_f^{\widetilde{\Gamma}}(Q_n \| P_{\theta,m}) \le D_f^{\widetilde{\Gamma}}(Q_n \| P_\theta) + \sup_{h \in \widetilde{\Gamma}, \theta \in \Theta} \{\Lambda_f^{P_\theta}[h] - \Lambda_f^{P_{\theta,m}}[h]\}. \tag{112}$$

Minimizing over $\theta \in \Theta$ and using the zero-approximation-error property we obtain

$$\inf_{\theta \in \Theta} D_f^{\widetilde{\Gamma}}(Q_n \| P_{\theta,m}) \le \inf_{\theta \in \Theta} D_f^{\widetilde{\Gamma}}(Q_n \| P_\theta) + \sup_{h \in \widetilde{\Gamma}, \theta \in \Theta} \{\Lambda_f^{P_\theta}[h] - \Lambda_f^{P_{\theta,m}}[h]\} \tag{113}$$

$$= \sup_{h \in \widetilde{\Gamma}, \theta \in \Theta} \{\Lambda_f^{P_\theta}[h] - \Lambda_f^{P_{\theta,m}}[h]\}.$$

Therefore

$$D_f^{\Gamma}(Q \| P_{\theta_{n,m}^*,m}) \le \sup_{h \in \widetilde{\Gamma}, \theta \in \Theta} \{\Lambda_f^{P_\theta}[h] - \Lambda_f^{P_{\theta,m}}[h]\} + \left(1 + (f^*)'_+(z_0 + \beta - \alpha)\right) \sup_{h \in \Gamma} \inf_{\tilde{h} \in \widetilde{\Gamma}} \|h - \tilde{h}\|_\infty + \epsilon_{n,m}^{opt} \tag{114}$$

$$+ D_f^{\widetilde{\Gamma}}(Q \| P_{\theta_{n,m}^*,m}) - D_f^{\widetilde{\Gamma}}(Q_n \| P_{\theta_{n,m}^*,m})$$

$$\le \sup_{h \in \widetilde{\Gamma}, \theta \in \Theta} \{\Lambda_f^{P_\theta}[h] - \Lambda_f^{P_{\theta,m}}[h]\} + \left(1 + (f^*)'_+(z_0 + \beta - \alpha)\right) \sup_{h \in \Gamma} \inf_{\tilde{h} \in \widetilde{\Gamma}} \|h - \tilde{h}\|_\infty + \epsilon_{n,m}^{opt}$$

$$+ \sup_{h \in \widetilde{\Gamma}} \{E_Q[h] - E_{Q_n}[h]\} + \sup_{h \in \widetilde{\Gamma}} \left\{\Lambda_f^{P_{\theta_{n,m}^*,m},m}[h] - \Lambda_f^{P_{\theta_{n,m}^*,m}}[h]\right\}.$$

Bounding the last term by maximizing over $\theta \in \Theta$ and then using $\Lambda_f^{P_\theta}[h] = \Lambda_f^{P_Z}[h \circ \Phi_\theta]$ and $\Lambda_f^{P_\theta,m}[h] = \Lambda_f^{P_Z,m}[h \circ \Phi_\theta]$ completes the proof. $\qquad\square$

## E  Concentration Inequalities for the Estimation of $(f, \Gamma)$-Divergences

The tools developed in Section 2 can also be used to address the simpler problem of concentration inequalities for the estimation of $D_f^\Gamma(Q\|P)$ using samples from $Q$ and $P$. Special cases of this problem were previously considered in Chen et al. (2023a; 2024). Both considered the case where $f$ corresponded to the family of $\alpha$-divergences and $\Gamma$ was the set of $L$-Lipschitz functions; the former focused on the consequences of group symmetry while the latter focused on heavy-tailed distributions. The techniques developed in this work allow us to derive concentration inequalities for a much wider class of $f$'s and $\Gamma$'s, though we do not consider the consequences of group symmetry in this work.

**Theorem E.1.** *Let $\Gamma$ be a nonempty set of measurable functions on $\mathcal{X}$ and suppose we have $\alpha, \beta \in \mathbb{R}$ such that $\alpha < \beta$ and $\alpha \le h \le \beta$ for all $h \in \Gamma$. Assume there exists a countable $\Gamma_0 \subset \Gamma$ such that for all $h \in \Gamma$ there exists a sequence $h_j \in \Gamma_0$ such that $h_j \to h$ pointwise. Let $f \in \mathcal{F}_1(a,b)$ with $a \ge 0$ and assume $f$ is strictly convex in a neighborhood of $1$ and $z_0 + \beta - \alpha \in \{f^* < \infty\}^o$.*

*Let $Q, P$ be probability measures on $\mathcal{X}$ and $X_i$, $i = 1, ..., n$, $X_i'$, $i = 1, ..., m$ be jointly independent samples from $Q$ and $P$ respectively with $Q_n, P_m$ the corresponding empirical measures. Then for $\epsilon > 0$ we have*

$$\mathbb{P}\left(D_f^\Gamma(Q\|P) - D_f^\Gamma(Q_n\|P_m) \ge \epsilon\right) \le \exp\left(-\frac{2\epsilon^2}{\frac{1}{n}(\beta - \alpha)^2 + \frac{1}{m}\Delta_{f,m}^2}\right), \tag{115}$$

$$\mathbb{P}\left(D_f^\Gamma(Q_n\|P_m) - D_f^\Gamma(Q\|P) \ge \epsilon + 2\mathcal{R}_{\Gamma,Q,n} + 2\mathcal{K}_{f,\Gamma,P,m}\right) \le \exp\left(-\frac{2\epsilon^2}{\frac{1}{n}(\beta - \alpha)^2 + \frac{1}{m}\Delta_{f,m}^2}\right), \tag{116}$$

*where $\Delta_{f,m}$ and $\mathcal{K}_{f,\Gamma,P,m}$ were defined in Definition 3.6.*

*Proof.* Similar to the proof of Theorem 3.7, if we define

$$H(X, X') = D_f^\Gamma(Q_n\|P_m), \tag{117}$$

then when $x$ and $\tilde{x}$ differ by a single component we have

$$|H(x, x') - H(\tilde{x}, x')| \leq \frac{1}{n}(\beta - \alpha) \tag{118}$$

and when $x'$ and $\tilde{x}'$ differ by a single component, Lemma 2.7 implies

$$|H(x, x') - H(x, \tilde{x}')| \leq \frac{1}{m}\Delta_{f,m}. \tag{119}$$

Therefore we can use McDiarmid's inequality to conclude that

$$\mathbb{P}\left(\pm(D_f^\Gamma(Q_n\|P_m) - \mathbb{E}[D_f^\Gamma(Q_n\|P_m)]) \geq \epsilon\right) \leq \exp\left(-\frac{2\epsilon^2}{\frac{1}{n}(\beta - \alpha)^2 + \frac{1}{m}\Delta_{f,m}^2}\right). \tag{120}$$

Combining (120) (lower sign) with the result of the calculation

$$
\begin{aligned}
\mathbb{E}\left[D_f^\Gamma(Q_n\|P_m)\right] &= \mathbb{E}\left[\sup_{h\in\Gamma, \nu\in[\alpha-z_0, \beta-z_0]} \{E_{Q_n}[h-\nu] - E_{P_m}[f^*(h-\nu)]\}\right] \\
&\geq \sup_{h\in\Gamma, \nu\in[\alpha-z_0, \beta-z_0]} \mathbb{E}\left[E_{Q_n}[h-\nu] - E_{P_m}[f^*(h-\nu)]\right] \\
&= \sup_{h\in\Gamma, \nu\in[\alpha-z_0, \beta-z_0]} \{E_Q[h-\nu] - E_P[f^*(h-\nu)]\} = D_f^\Gamma(Q\|P)
\end{aligned}
\tag{121}
$$

we obtain (115). Combining (120) (upper sign) with the result of the calculation

$$\mathbb{E}\left[D_f^\Gamma(Q_n\|P_m)\right] - D_f^\Gamma(Q\|P) \leq \mathbb{E}\left[\sup_{h\in\Gamma}\{E_{Q_n}[h] - E_Q[h]\}\right] + \mathbb{E}\left[\sup_{h\in\Gamma}\left\{\Lambda_f^P[h] - \Lambda_f^{P_m}[h]\right\}\right] \tag{122}$$

and then using Theorem C.4 and Lemmas 2.9 and 2.10 we obtain (116). $\qquad\square$

## F Concentration Inequalities for Reverse $(f, \Gamma)$-GANs

The asymmetry of the $(f, \Gamma)$-divergences requires us to separately treat the cases where the generator is the first argument and where it is the second; we emphasize that this choice can make a significant difference in practice as shown in the examples in Birrell et al. (2022b). With this in mind we give a second version of the $(f, \Gamma)$-GAN assumptions. The proofs for this case are very similar and so we omit them.

**Assumption F.1** (Reverse $(f, \Gamma)$-GAN Assumptions). *Assume that Assumption 3.2 holds, with the sole exception being that $\theta_{n,m}^*$ satisfies*

$$D_f^{\widetilde{\Gamma}}(P_{\theta_{n,m}^*, m}\|Q_n) \leq \inf_{\theta\in\Theta} D_f^{\widetilde{\Gamma}}(P_{\theta, m}\|Q_n) + \epsilon_{n,m}^{opt} \quad \mathbb{P}\text{-}a.s. \tag{123}$$

*as opposed to (23).*

**Lemma F.2** (Reverse $(f, \Gamma)$-GAN Error Decomposition). *Under Assumption F.1 the $(f, \Gamma)$-GAN error can be decomposed $\mathbb{P}$-a.s. as follows:*

$$
\begin{aligned}
&D_f^\Gamma(P_{\theta_{n,m}^*}\|Q) - \inf_{\theta\in\Theta} D_f^\Gamma(P_\theta\|Q) \\
&\leq \sup_{h\in\widetilde{\Gamma}, \theta\in\Theta}\left\{E_{P_Z}[h\circ\Phi_\theta] - E_{P_{Z,m}}[h\circ\Phi_\theta]\right\} + \sup_{h\in\widetilde{\Gamma}, \theta\in\Theta}\left\{E_{P_{Z,m}}[h\circ\Phi_\theta] - E_{P_Z}[h\circ\Phi_\theta]\right\} \\
&\quad + \sup_{h\in\widetilde{\Gamma}}\left\{\Lambda_f^{Q_n}[h] - \Lambda_f^Q[h]\right\} + \sup_{h\in\widetilde{\Gamma}}\left\{\Lambda_f^Q[h] - \Lambda_f^{Q_n}[h]\right\} \\
&\quad + \left(1 + (f^*)'_+(z_0 + \beta - \alpha)\right)\sup_{h\in\Gamma}\inf_{\tilde{h}\in\widetilde{\Gamma}}\|h - \tilde{h}\|_\infty + \epsilon_{n,m}^{opt}.
\end{aligned}
\tag{124}
$$

**Lemma F.3** (Reverse $(f, \Gamma)$-GAN Error Decomposition 2). *Under Assumption F.1, and supposing that* $\inf_{\theta \in \Theta} D_f^{\widetilde{\Gamma}}(P_\theta \| \mu_n) = 0$ *for all empirical distributions* $\mu_n$*, the* $(f, \Gamma)$-GAN error can be decomposed $\mathbb{P}$-a.s. as follows:

$$D_f^\Gamma(P_{\theta_{n,m}^*} \| Q) \leq \sup_{h \in \widetilde{\Gamma}, \theta \in \Theta} \left\{ E_{P_Z}[h \circ \Phi_\theta] - E_{P_{Z,m}}[h \circ \Phi_\theta] \right\} + \sup_{h \in \widetilde{\Gamma}, \theta \in \Theta} \left\{ E_{P_{Z,m}}[h \circ \Phi_\theta] - E_{P_Z}[h \circ \Phi_\theta] \right\} \quad (125)$$
$$+ \sup_{h \in \widetilde{\Gamma}} \left\{ \Lambda_f^{Q_n}[h] - \Lambda_f^Q[h] \right\} + \left( 1 + (f^*)_+'(z_0 + \beta - \alpha) \right) \sup_{h \in \Gamma} \inf_{\tilde{h} \in \widetilde{\Gamma}} \| h - \tilde{h} \|_\infty + \epsilon_{n,m}^{opt}.$$

**Theorem F.4** (Reverse $(f, \Gamma)$-GAN Concentration Inequalities). *Under Assumption F.1, and in particular with* $\theta_{n,m}^*$ *the approximate solution to the empirical* $(f, \Gamma)$-GAN problem (123), for $\epsilon > 0$ we have

$$\mathbb{P}\left( D_f^\Gamma(P_{\theta_{n,m}^*} \| Q) - \inf_{\theta \in \Theta} D_f^\Gamma(P_\theta \| Q) \geq \epsilon + \epsilon_{approx}^{\Gamma, \widetilde{\Gamma}} + \epsilon_{opt}^{n,m} + 4\mathcal{R}_{\widetilde{\Gamma} \circ \Phi, P_Z, m} + 4\mathcal{K}_{f, \widetilde{\Gamma}, Q, n} \right) \quad (126)$$
$$\leq \exp\left( -\frac{\epsilon^2}{\frac{2}{m}(\beta - \alpha)^2 + \frac{2}{n}\Delta_{f,n}^2} \right),$$

*where we refer to the quantities in Definition 3.6.*

*If, in addition,* $\inf_{\theta \in \Theta} D_f^{\widetilde{\Gamma}}(P_\theta \| \mu_n) = 0$ *for all possible empirical distributions* $\mu_n$ *then we obtain the tighter bound*

$$\mathbb{P}\left( D_f^\Gamma(P_{\theta_{n,m}^*} \| Q) \geq \epsilon + \epsilon_{approx}^{\Gamma, \widetilde{\Gamma}} + \epsilon_{opt}^{n,m} + 4\mathcal{R}_{\widetilde{\Gamma} \circ \Phi, P_Z, m} + 2\mathcal{K}_{f, \widetilde{\Gamma}, Q, n} \right) \leq \exp\left( -\frac{\epsilon^2}{\frac{2}{m}(\beta - \alpha)^2 + \frac{1}{2n}\Delta_{f,n}^2} \right). \quad (127)$$

