# OpenReview forum: "Statistical Error Bounds for GANs with Nonlinear Objective Functionals"
_TMLR — Accepted by TMLR_

### Review · Reviewer_NE8H · 2025-02-21

**Summary Of Contributions:**

The paper studies generative adversarial network (GAN) formulations formulated using a constrained discriminator set with an $f$-divergence loss function. This family of GANs, which the authors define as $(f,\Gamma)$-GANs, has been often used in the literature. The paper shows a statistical consistency result aimed at $(f,\Gamma)$-GANs. Theorem 3.7 bounds the gap between the $(f,\Gamma)$-divergence of the optimized generator $\theta^*_{n,m}$ and the infimum achievable divergence by assuming bounded discriminator approximation and optimization error, and the Rademacher complexity of the discriminator set.

**Audience:**

Yes

**Claims And Evidence:**

Yes

**Requested Changes:**

The following would make the paper stronger:

1. Discussing the application of the result to standard choices of $f$-divergence measures, such as JS and KL divergence. Especially, it would help to show a particular choice of $f$ might be preferred based on the paper's theorems.

2. Adding some numerical results on synthetic and perhaps image data that examine or support the main theorem.

3. Including more discussion on the utility of the theoretical results and their implication on the design and performance of GANs in practice.

**Strengths And Weaknesses:**

**Strengths**

1. The paper studies the theory of GANs, which is a highly interesting subject.
2. The paper presents an error bound for $(f,\Gamma)$-GANs with a non-linear function $f$, an interesting result that applies to several successful GAN formulations.

**Weaknesses**

1. Although I find the paper's main result in Theorem 3.7 interesting, I am unsure whether this result is sufficient for an independent publication at the level of TMLR. While it is nice to have a separate bound for the authors' defined $(f,\Gamma)$-divergence, the result appears to only address the case of a non-linear $f$. Furthermore, the paper lacks bounds or novel discussion about the main error components that remain in the bound: 1) the approximation error term, 2) the optimization error term, and 3) the Rademacher complexity. These generally defined error terms are left unanalyzed, and in my opinion, Theorem 3.7 alone is not sufficient for a top-tier paper on the theory of GANs.
2. To elaborate on my previous comment regarding the paper's result, let's discuss the error terms in Theorem 3.7. The approximation error in equation (30) includes the term $\sup_{h\in\Gamma} \inf_{\tilde{h}\in\tilde{\Gamma}} || h - \tilde{h} ||_{\infty}$. I am concerned that this term may not be small for standard choices where $\Gamma$ is the set of 1-Lipschitz functions and $\tilde{\Gamma}$ is a set of neural networks. For this error term to be bounded by $\epsilon$, we would require an everywhere $\epsilon$-accurate approximation of every 1-Lipschitz function, which seems highly unlikely in moderately high dimensions. As the authors do not analyze this error term, it is unclear if such a bound is useful for the discussed case where $\Gamma$ is the set of 1-Lipschitz functions.
3. Similar to the approximation error term, there is no discussion on the optimization error term, which is indeed a critical challenge in training GANs. Additionally, the paper uses a standard uniform convergence analysis for bounding the estimation error, based on a general Rademacher complexity term, which would likely lead to a vacuous bound for a standard deep neural network discriminator. Including some discussion on the applicability of the result to standard GANs would address this concern.
4. Based on the above, the paper's contribution appears limited to deriving the additional constants in the bound for a non-linear $f$-divergence function $f$, assuming the scalar input to $f$ varies within a limited range $[\alpha,\beta]$. To demonstrate the utility of this result, it would be beneficial to discuss its implications for standard $f$-divergence cases such as JS or KL divergences. Currently, the paper presents the result for a general $f$, leaving the potential value of the result in determining which function $f$ to use unclear.
5. The paper lacks numerical results. While running image-scale GAN experiments may be challenging, synthetic experiments with uniform or Gaussian data distributions would be valuable.
6. The paper's literature review misses several related works on the theoretical analysis of $f$-GAN and IPM-GANs. The following papers also theoretically study $f$-GANs and IPM-GANs with restricted discriminator sets:
    * Arora et al, "Generalization and Equilibrium in Generative Adversarial Nets", ICML 2017
    * Liu et al, "Approximation and Convergence Properties of Generative Adversarial Learning", NeurIPS 2017
    * Farnia et al, "A Convex Duality Framework for GANs", NeurIPS 2018
    * Liu et al, "The Inductive Bias of Restricted f-GANs"

---

> ### Author Response · Authors · 2025-03-28
> **Response to weaknesses and requested changes**
>
> Thank you for your comments and suggestions. To address your requested changes, we added Table 1 on page 9, which gives several important examples to which our theory applies.  Regarding numerical results, versions of the GANs discussed here have already been studied empirically in references such as Nowozin et al. 2016 and Birrell et al. 2022b. Thus we do not believe it necessary to include additional experiments here. However, we did summarize some of the most important findings at the end of the first paragraph on page 2 in order to better motivate our study in this work. We also added a paragraph above remark 3.9 on page 13 which discusses some additional messages one can take from our theorems.
>
> Below we give additional responses to several of the numbered weaknesses, including some additional changes we made to address them:
>
> Weakness 1:  We wish to clarify that the results do not require that $f^*$ not be linear, rather we generalize the analysis to also cover the nonlinear case. Our results (continuously) reduce to and extend the known results in the linear case (i.e., the IPM case). We clarify this in the new paragraph added prior to Remark 3.9 on page 13.
>
> Weakness 2:  We have added additional discussion of the approximation error to the paragraph after (9). In particular, we emphasize that, while we stated the result for the general case, one can always choose $\widetilde{\Gamma}=\Gamma$ and thus make the approximation error equal to zero (i.e., use a NN divergence). Apart from the $f^*$-dependent prefactor, the form of the approximation error when $\widetilde{\Gamma}\not=\Gamma$ is not unique to our work and has been studied elsewhere in Huang et al. 2022. Thus we consider further discussion of this to be our of the scope of the current work.
>
> Weakness 3: As for the Rademacher complexity, we did provide a detailed study of this in our Section 3.3.  While our use of Rademacher complexity to obtain ULLN bounds is a standard technique, our method of bounding the Rademacher complexities themselves is, to our knowledge, novel and provides new insight into both the case of general nonlinear $f^*$ as well as the linear IPM GAN case. Specifically, our Theorem 3.12 allows our Theorem 3.7 to be applied to distributions with non-compact support, through a sample-dependent local Lipschitz constant $L(z)$ that is only required to have finite second moment.  For many standard NN architectures, $L(z)$ is linearly bounded, hence our method gives non-vacuous (finite) results even for many heavy tailed cases (e.g., the MGF does not exist but the second moment is finite).   Thus, for  NN divergences (i.e., when $\widetilde{\Gamma}=\Gamma$) the only unexplored term in our bounds is the optimization error, with all other terms explicitly finite. While we certainly agree that the optimization error is important, it depends on the details of the optimization algorithm used and we consider that to be a separate problem that is out of the current scope.  Thus we have taken the same route as, to our knowledge, all existing statistical error analyses of GANs, including Arora et al. 2017 and the cited studies of IPM GANs, and simply included the optimization error as a separate term in the bound.
>
> Weakness 6: Thank you for pointing out these references.  We added them to the background discussion, starting at the bottom of page 1. Of these, the only one which studies statistical error is Arora et al. 2017; specifically they studied GANs with objectives that are patterned after the original (JS) GAN, which have a different nature from the $(f,\Gamma)$-GANs studied here.  We did add a more detailed comparison of Arora et al. 2017 with our work to the new Section 1.2 on pages 4-5.

---

### Review · Reviewer_GdXz · 2025-03-08

**Summary Of Contributions:**

- The primary contribution is the derivation of finite-sample concentration inequalities for GANs, establishing theoretical guarantees on the statistical consistency of $(f,\Gamma)$ GANs.
- This paper uses new techniques to handle the nonlinearity of the objective function introduced by the generalized cumulant generating function. These methods hold potential applicability beyond GANs.
- The authors derive the improved Rademacher complexity bounds under less restrictive distributional assumptions, making the analysis more closely aligned with practical scenarios.
- The derived bounds generalize previously known results for IPM-GANs, recovering them explicitly as a special case when $f^*=z$.

**Audience:**

Yes

**Claims And Evidence:**

Yes

**Requested Changes:**

- Minors:
    - Currently, citations appear as inline text, which can disrupt readability. Using parenthetical citations when it is not part of the sentence would be better.
    - Define $m,n$ when they are first introduced in Section 1.1. Their definitions weren’t clear to me until I saw Lemma 2.5.

**Strengths And Weaknesses:**

**Strengths**

 - The paper is structured logically, with clear definitions, theorems, and remarks, helping understand of the complex theoretical arguments. The manuscript is enjoyable and instructive to read.
- Rigorous and comprehensive theoretical analysis to prove the statistical consistency for the general class of GANs. Compared to the exiting literature that focuses on IPM-GAN with linear objectives, this paper extends further towards more general (f,\Gamma) GANs with notably weaker assumptions.

**Weaknesses**

- While theoretically strong, the paper currently offers limited direct practical guidance for designing and training GANs.

---

> ### Author Response · Authors · 2025-03-28
> **Response to weaknesses and requested changes**
>
> Thank you for your comments and suggestions. We have added the definitions of n and m to the start of Section 1.1 and have changed citations to parenthetical where appropriate.  We agree that our results don't necessarily provide insight into designing GANs. Rather we view our results as justifying, from a statistical perspective, the use of $(f,\Gamma)$-GANs as an extended family of methods that naturally generalize IPM-GANs such as WGAN.   Such an extended family is useful because the nonlinear nature of the objective functional in $(f,\Gamma)$-GANs was previously observed in Birrell et al. (2022b) to provide practical advantages over WGAN.  Specifically: 1)  it was found that the nonlinear objective makes the training less sensitive to hyperparameter tuning, 2) $(f,\Gamma)$-GANs are applicable to a much wider variety of heavy tailed distributions than WGAN.  To better motivate our study, we have added a few sentences on this to the end of the first paragraph on page 2.

---

### Review · Reviewer_xG3i · 2025-03-26

**Summary Of Contributions:**

This paper considers a class of GAN-style objectives that interpolates between $f$-divergences and integral probability metrics, previously introduced in work of Birrell et al. '22. They key difference between this and typical IPM-style GANs like Wasserstein GAN is that the objective depends in a nonlinear fashion on the discriminator. The main finding of this work is a generalization bound: if one tries to minimize this measure of discrepancy between {empirical distribution over i.i.d. samples from the target distribution} and any {generator that pushes forward an empirical distribution over i.i.d. samples from the seed distribution}, the discrepancy between the target distribution and the pushforward of the seed distribution under the resulting generator can be controlled by the usual three types of terms: finite-sample error, optimization error, and approximation error term. When the space of discriminators has bounded range, this can be further strengthened to a high-probability bound.

**Audience:**

No

**Broader Impact Concerns:**

N/A: while the paper is on generative modeling, it is a theory paper proving statistical error bounds and there are no relevant ethical implications

**Claims And Evidence:**

Yes

**Requested Changes:**

While the paper is technically sound, I would have liked further discussion on the motivation for this paper and for $(f,\Gamma)$-divergences. While these are a generalization of existing GAN approaches, what is the advantage of this formulation over previous ones, e.g. with regards to training stability and mode collapse? I would have imagined that the more complicated objective makes these even more finicky to train. Given the decline in popularity of this method for generative modeling, it is especially important that papers in this space try to offer counterpoints that make a case for this approach or variants thereof.

**Strengths And Weaknesses:**

Strengths:
- The nonlinear dependence on the test function introduces various technical hurdles which make it more challenging to establish properties that would otherwise be straightforward in the IPM setting, e.g. the sensitivity bound in Lemma 2.7 where one shows that the generalized cgf for an empirical distribution does not change too much when one data point is changed.
- The main conceptual contribution is that this consistency result allows one to interpolate between existing consistency results for GANs based on IPMs and GANs based on $f$-divergences.
- They establish finite-sample concentration bounds (the aforementioned Lemma 2.7 is crucial here)

Weaknesses:
- As with any paper proving generalization bounds, there is the drawback that it does not touch upon optimization aspects
- My main reservation about this work is the motivation, see Requested Changes below

---

> ### Author Response · Authors · 2025-03-28
> **Response to weaknesses and requested changes**
>
> Thank you for your comments and suggestions.  Our study of the statistical properties of $(f,\Gamma)$-GANs is motivated by the fact that the nonlinear nature of their objective functional was previously observed in Birrell et al. (2022b) to provide practical advantages over linear objectives like in WGAN.  In particular:
>
> 1) Rather than being destabilizing, it was found that the nonlinear objective makes the training less sensitive to hyperparameter tuning.
> 2) $(f,\Gamma)$-GANs are applicable to heavy tailed distributions; specifically, they allow for much heavier tails than can be treated by WGAN.
>
> To better motivate our study of the statistical properties of $(f,\Gamma)$-GANs, we have added few sentences regarding these advantages at the end of the first paragraph on page 2.

---

### Decision · Action_Editor_Je7M · 2025-04-30

**Recommendation:** Accept as is

**Comment:**

This work provided a theoretical analysis on the statistical error bound of a family of generative adversarial networks based on the so-called $(f, \Gamma)$-divergence (an interpolation between the $f$-divergence and the integral probability metric, IPM). The main technical challenge is to deal with the nonlinear objective (as opposed to the linear one in IPMs) due to explicitly accounting for translation invariance. The paper is expertly written and the results seem to strictly extend existing analysis.

One one hand, all reviewers recognized the technical contributions, including the presentation and soundness. On the other hand, the reviewers voiced concerns on the applicability and relevance of the theoretical analysis to existing practices. In the response, the author further explained how the current results extend previous ones and pointed out the value of this work: more on justifying some of the existing practices than motivating new designs, which I tend to agree.

In the end, I believe the contributions of this work outweigh the (practical) concerns: while it may not (directly) inform practitioners how to train GANs better, it is nevertheless assuring to know when some of the GAN variants are statistically sound (not to mention that some of the results may be of independent interest).

**Audience:**

Yes.

**Claims And Evidence:**

Yes: the results are rigorously proved and explained.